# Loss of neuronal Miro1 disrupts mitophagy and induces hyperactivation of the integrated stress response

Guillermo López-Doménech[1,†] (iD), Jack H Howden[1,†] (iD), Christian Covill-Cooke[1], Corinne Morfill[1], Jigna V Patel[2], Roland Bürli[3], Damian Crowther[3] (iD), Nicol Birsa[4] (iD), Nicholas J Brandon[5] & Josef T Kittler[1,*] (iD)

## Abstract

Clearance of mitochondria following damage is critical for neuronal homeostasis. Here, we investigate the role of Miro proteins in mitochondrial turnover by the PINK1/Parkin mitochondrial quality control system *in vitro* and *in vivo*. We find that upon mitochondrial damage, Miro is promiscuously ubiquitinated on multiple lysine residues. Genetic deletion of Miro or block of Miro1 ubiquitination and subsequent degradation lead to delayed translocation of the E3 ubiquitin ligase Parkin onto damaged mitochondria and reduced mitochondrial clearance in both fibroblasts and cultured neurons. Disrupted mitophagy *in vivo*, upon postnatal knockout of Miro1 in hippocampus and cortex, leads to a dramatic increase in mitofusin levels, the appearance of enlarged and hyperfused mitochondria and hyperactivation of the integrated stress response (ISR). Altogether, our results provide new insights into the central role of Miro1 in the regulation of mitochondrial homeostasis and further implicate Miro1 dysfunction in the pathogenesis of human neurodegenerative disease.

**Keywords** eIF2α; megamitochondria; Parkinson's disease; Rhot1; Rhot2
**Subject Categories** Autophagy & Cell Death; Membranes & Trafficking; Post-translational Modifications & Proteolysis
**The EMBO Journal (2021) 40: e100715**

## Introduction

Mitochondria are double-membraned organelles essential for a wide range of metabolic processes. Mitochondria are the main source of cellular ATP, are critical in maintaining $Ca^{2+}$ homeostasis and are home to lipid biosynthetic pathways. Due to their normal activity, mitochondria eventually become less efficient and a source of reactive oxygen species (ROS). When this happens, cells have in place mitochondrial quality control mechanisms to identify, isolate and remove dysfunctional mitochondria (Pickles *et al*, 2018). PINK1 (PTEN-induced putative kinase 1; *PARK6*), a mitochondrial serine–threonine kinase, and Parkin (*PARK2*), an E3 ubiquitin ligase, are components of a mitochondrial quality control apparatus that promotes the selective turnover of damaged mitochondria through mitochondrial autophagy (mitophagy). PINK1, normally imported into the mitochondrion and cleaved at the inner mitochondrial membrane (IMM), selectively accumulates in its full-length form on the outer mitochondrial membrane (OMM) of damaged mitochondria, where it phosphorylates the serine 65 (S65) residue of ubiquitin, as well as a conserved residue in the ubiquitin-like domain of Parkin. This leads to the recruitment and activation of Parkin from the cytosol to the mitochondria to ubiquitinate various OMM substrates (Vives-Bauza *et al*, 2010; Deas *et al*, 2011; Cai *et al*, 2012; Exner *et al*, 2012; Sarraf *et al*, 2013; Lazarou *et al*, 2015). The ubiquitination of Parkin substrates on the OMM triggers the recruitment of autophagic adaptors (e.g. p62, NDP52, optineurin) and is a crucial step in the clearance of damaged mitochondria through the autophagic pathway (Narendra *et al*, 2010a; Lazarou *et al*, 2015). Importantly, loss-of-function mutations in PINK1 and Parkin are associated with rare recessive forms of Parkinson's disease (PD) (Thomas & Beal, 2007) supporting an important role for mitophagy in neuronal survival and a link between its dysregulation and neurodegenerative diseases.

A key target of Parkin-mediated ubiquitination is the Miro (mitochondrial Rho) family of GTPases. In mammals, there are two members of the family, Miro1 and Miro2, which contain two GTPase domains flanking two $Ca^{2+}$-sensing EF-hand motifs and a transmembrane domain that anchors them to the OMM (Fransson

1 Neuroscience, Physiology and Pharmacology, University College London, London, UK
2 MRC Laboratory for Molecular Cell Biology, University College London, London, UK
3 Neuroscience, IMED Biotech Unit, AstraZeneca, Cambridge, UK
4 UCL Institute of Neurology, Queen Square, London, UK
5 Neuroscience, IMED Biotech Unit, AstraZeneca, Boston, MA, USA
 *Corresponding author. Tel: +44 20 7679 3218; E-mail: j.kittler@ucl.ac.uk
 †These authors contributed equally to this work

*et al*, 2003). Miro proteins have emerged as critical regulators of mitochondrial trafficking and distribution (Guo *et al*, 2005; Fransson *et al*, 2006; Lopez-Domenech *et al*, 2016) and may also have other important roles for mitochondrial function as components of mitochondria-ER contact sites and as regulators of mitochondrial calcium homeostasis (Kornmann *et al*, 2011; Lee *et al*, 2016; Niescier *et al*, 2018; Modi *et al*, 2019). Under mitochondrial damage, Miro proteins are rapidly ubiquitinated and degraded by a PINK1/Parkin-dependent mechanism (Wang *et al*, 2011b; Liu *et al*, 2012; Birsa *et al*, 2014; Ordureau *et al*, 2018). Regulation of the Miro trafficking complex by PINK1 and Parkin may serve to dissociate damaged mitochondria from the microtubule and actin transport pathways (Wang *et al*, 2011b; Lopez-Domenech *et al*, 2018), helping to isolate the damaged organelles from the functional mitochondrial network. We and others have previously reported that, in addition to acting as a Parkin substrate, Miro proteins might directly act as receptors for Parkin on the OMM to facilitate Parkin recruitment and stabilisation on the OMM (Birsa *et al*, 2014; Shlevkov *et al*, 2016; Safiulina *et al*, 2019). Furthermore, Miro degradation was shown to be impaired in PD patients' derived fibroblasts (Hsieh *et al*, 2016) and several Miro1 mutations have been identified as risk factors in patients with PD, further supporting the involvement of Miro1 in the pathogenesis of the disease (Berenguer-Escuder *et al*, 2019; Grossmann *et al*, 2019; Berenguer-Escuder *et al*, 2020). Despite the emerging role for Miro in the regulation of mitophagy and the growing links between Miro1 and human PD pathology, the long-term consequences of Miro1 disruption for mitochondrial and neuronal homeostasis *in vivo* remain very poorly understood.

Here, by using Miro knockout mouse embryonic fibroblasts (MEFs) and cultured hippocampal neurons, we show that Miro1 is required for the efficient stabilisation of Parkin on the OMM following mitochondrial damage. In addition, blocking Miro1 ubiquitination stabilises Miro levels upon mitochondrial damage and also leads to slowed mitophagy suggesting that a tight temporal regulation of Miro1 levels at the OMM is required for efficient mitophagy. *In vivo*, disruption of mitophagy in post-natal forebrain neurons in conditional Miro1 knockout brains (Miro1^CKO) is associated with age-dependent increase of the mitochondrial fusion proteins Mfn1 and Mfn2, mitochondrial hyperfusion and pathological hyperactivation of the integrated stress response (ISR). Our results provide new insights into the role of Miro1 in the regulation of the PINK1/Parkin-dependent mitophagy and the consequences of its dysregulation for neurological disease.

## Results

### Damage-induced mitophagy is slowed in Miro^DKO cells

While it is well established that Miro proteins are rapidly ubiquitinated by Parkin upon mitochondrial damage (Liu *et al*, 2012; Birsa *et al*, 2014), reciprocally, how important Miro ubiquitination may be for the mitophagic process remains less clear. We previously reported that Miro itself might act as a stabiliser of mitochondrial Parkin, acting in the first steps of Parkin-dependent mitophagy (Birsa *et al*, 2014). To further investigate the importance of Miro for damage-induced Parkin stabilisation, we used a recently characterised Miro1/2 double-knockout (Miro^DKO) MEF cell line (Lopez-

Domenech *et al*, 2018). Expressing ^YFPParkin and inducing mitochondrial damage with FCCP is a well-characterised assay for studying the time-dependent translocation of Parkin onto the mitochondria. Using this assay, we followed 4 key stages of Parkin distribution and mitochondrial remodelling during FCCP-induced mitochondrial damage (Fig 1A): (i) diffuse Parkin distribution within cells; (ii) appearance of sparse Parkin puncta onto mitochondrial units; (iii) mitochondrial aggregation and perinuclear redistribution of Parkin positive mitochondria; and (iv) complete translocation of Parkin onto all the mitochondrial network. Blind scoring allowed us to compare the above stages of the mitophagic process between wild-type (WT) control and Miro^DKO MEFs. At early time points, 1 and 3 h after FCCP treatment, we observed a significantly higher proportion of Miro^DKO cells still presenting a diffuse distribution of Parkin compared to WT cells (Fig 1B). This correlated with a delay in the appearance of Parkin positive puncta which peaked at 1 h of FCCP treatment in WT cells but only between 3 and 6 h in Miro^DKO cells (Fig 1C). Parkin translocation soon after mitochondrial damage induces the remodelling and perinuclear aggregation of mitochondria which was also significantly delayed in Miro^DKO MEFs (Fig 1D) at 3 h, compared to WT cells. Finally, by 6 h, over 50% of WT cells exhibited an almost complete Parkin translocation onto the mitochondrial network, which was greatly reduced in the Miro^DKO MEFs (Fig 1E). The delay in Parkin translocation to damaged mitochondria is not due to reduced Myo19 levels – which we recently showed are decreased in Miro^DKO cells – as both WT and Myo19^KO cells showed similar dynamics of Parkin translocation after FCCP treatment (Appendix Fig S1A and B). Moreover, Miro deletion did not affect the timely stabilisation of PINK1 at very early stages of the mitophagic process (15–60 min after FCCP treatment) (Fig 1F and G) indicating that the presence of Miro at the OMM appears to be critical for the efficient recruitment of Parkin to damaged mitochondria and for the progression of mitophagy.

### Miro1 is promiscuously ubiquitinated on multiple lysine residues

To further understand the role of Miro in mitophagy and mitochondrial clearance, we set out to further characterise lysine residues important for Miro1 ubiquitination in cells. Several lysine residues have been identified as potential sites for Miro1 ubiquitination, both from *in vitro* studies (Kazlauskaite *et al*, 2014) and from unbiased mass spectrometric approaches (Table S1). We generated a series of myc-tagged single point mutants in Miro1 to some of these sites along with a compound mutation lacking five key potential sites (Miro1^5R hereafter). We selected the residues K153, K187 and K572 because they were identified in several studies as being ubiquitinated upon mitochondrial damage, and K182 and K194 due to their close proximity to K187, likely a critical ubiquitination target (Fig 2A and Appendix Table S1). In addition, we synthesised a human Miro1 mutant construct in which all lysine residues (except K612 and K616 to avoid mistargeted localisation) were replaced by arginine (Miro1^allR). Because lysine 572 (K572) was previously demonstrated to be a key site for Parkin-dependent ubiquitination *in vitro* (Kazlauskaite *et al*, 2014; Klosowiak *et al*, 2016), we also generated an additional construct where this residue was reintroduced on the Miro1^allR backbone (Miro1^R572K). All constructs were effectively targeted to mitochondria and expressed at mostly comparable levels in SH-SY5Y cells stably overexpressing ^FlagParkin, where

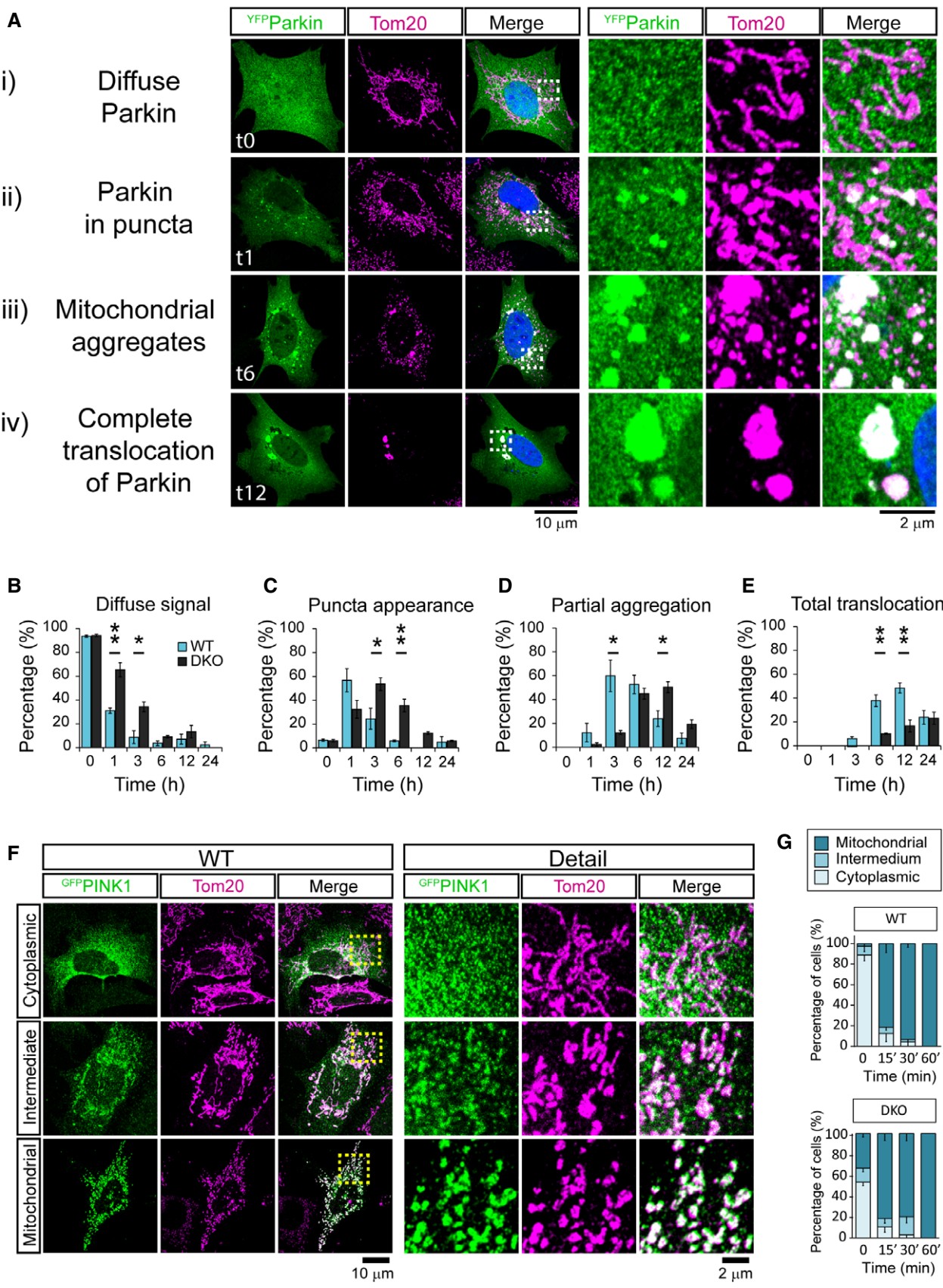

**Figure 1.**

◄

**Figure 1.   Delayed Parkin recruitment to mitochondria in Miro$^{DKO}$ cells upon mitochondrial damage.**

A–E   (A) Representative images (and details from boxed regions) of WT MEF cells transfected with $^{YFP}$Parkin (green) and treated with FCCP (10 μM) to induce mitophagy. Cells were immunostained with anti-Tom20 (magenta) to reveal the mitochondrial network. Each row represents one stage in the process of mitophagy that has been used to score the progression of mitophagy at different time points for the different conditions tested (the specific time point of each example is indicated in the image). From top to bottom: (i) "Diffuse Parkin" defines cells where Parkin expression is homogenously distributed throughout the cytoplasm (quantified in B). (ii) Soon after mitochondrial damage, Parkin appears enriched in small "puncta" throughout the cytoplasm localising within isolated mitochondria (quantified in C). (iii) Later in the process, these puncta aggregate into bigger structures accumulating in perinuclear regions (quantified in D). (iv) The process of Parkin translocation eventually affects all the mitochondrial population which usually collapses into several large structures possibly forming part of big autolysosomes in the perinuclear region (quantified in D). Scoring was performed from 3 independent experiments (*n* = 3; one-way ANOVA with Dunnett post-test).

F, G   (F) Representative images (and details from boxed regions) of WT MEF cells transfected with $^{GFP}$PINK1 (green) and treated with FCCP (10 μM) to induce mitophagy. Cells were immunostained with anti-Tom20 (magenta) to reveal the mitochondrial network. Each row represents one of the three categories of GFP signal used to describe PINK1 stabilisation on mitochondria after mitochondrial damage (cytoplasmic, intermediate and mitochondrial). The scoring of these categories was used to describe PINK1 stabilisation on mitochondria in WT and Miro$^{DKO}$ cells at different time points shortly after mitochondrial insult (G). Data collected from three independent experiments (*n* = 3).

Data information: Error bars represent SEM. Significance: *$P$ < 0.05 and **$P$ < 0.01.

robust damage-induced Miro1 ubiquitination has been previously reported (Birsa *et al*, 2014) (Fig 2B and C and Appendix Fig S2A). We then tested the ability of these constructs to be ubiquitinated by inducing mitochondrial damage. Surprisingly, the compound mutation of the 5 key potential sites of Miro ubiquitination in the Miro1$^{5R}$ construct led to a significant (but only partial) decrease in the levels of damage-induced Miro1 ubiquitination (Fig 2D and E) suggesting that in cells, as opposed to *in vitro*, multiple lysine residues may serve as substrates for Miro1 ubiquitination. Only by mutating all potential Miro1 ubiquitination sites could we completely block its damage-induced ubiquitination (Fig 2D and E); thus, Parkin exhibits significant promiscuity in targeting lysine residues on Miro1. In agreement with the conclusion that no individual site is necessary and sufficient to rescue Miro1 ubiquitination, reintroducing K572 on the Miro1$^{allR}$ background was unable to rescue the extent of damage-induced Miro1 ubiquitination (Fig 2D and E). Interestingly, under mitochondrial damage, defects in Miro1 ubiquitination correlated with the apparent stabilisation of Miro1 mutants (Fig 2D and Appendix Fig S2B). To further investigate the impact on time-dependent FCCP-induced Miro1 loss, we then expressed wild-type Miro1 (Miro1$^{WT}$), Miro1$^{5R}$ or Miro1$^{allR}$ and treated the cells with FCCP at different time points. Blocking Miro1 ubiquitination by mutating all lysines in Miro1 results in the complete stabilisation of Miro1 protein levels and blockade of damage-induced Miro1 degradation at both 3 and 6 h time points. This stabilisation does not occur when Miro1 ubiquitination is only reduced in the Miro1$^{5R}$ condition (Fig 2F and G) at the time points studied. Unexpectedly, however, we also noted that the damage-induced loss of PDH-E1α (a mitochondrial matrix protein) was reduced in both Miro1$^{5R}$ and Miro1$^{allR}$ expressing cells after 3 h of FCCP treatment (Fig 2H), while the non-ubiquitinated form of Parkin also appeared to be stabilised at this time point (Fig 2I). This suggested that reducing Miro1 ubiquitination may impact on the rate of turnover of damaged mitochondria and that Miro ubiquitination and degradation may not just be required to stop the trafficking of damaged mitochondria but may be directly involved in the mitophagic process.

## Ubiquitination and degradation of Miro1 are required steps for the induction and progression of mitophagy, respectively

To further investigate mechanistically the importance and specificity of Miro1 ubiquitination for Parkin translocation and mitochondrial clearance, we performed rescue experiments in Miro$^{DKO}$ MEFs with Miro1$^{WT}$, the ubiquitination mutants Miro1$^{5R}$ and Miro1$^{allR}$ as well as with a $^{myc}$Miro2 construct to test the specificity of Miro regulation of mitophagy. None of the constructs used altered the basal membrane potential of mitochondria when compared with untransfected cells in the same cultures ruling out an upstream effect on mitochondrial homeostasis (Appendix Fig S3A and B). All key stages of damage-induced Parkin redistribution and mitochondrial remodelling were then examined. At early time points (1–3 h), only re-expression of Miro1, but not Miro2, rescued the proportion of cells at advanced stages of mitophagy (cells showing Parkin-induced mitochondrial aggregation and perinuclear redistribution of mitochondria) (Fig 3A–C) suggesting that Miro1, but not Miro2, is the main regulator of mitophagy. Furthermore, none of the ubiquitin mutants were able to rescue mitophagy progression similar to WT Miro1 (Fig 3A–C) indicating that the correct ubiquitination of Miro1 is critical to efficiently recruit Parkin to damaged mitochondria. Importantly, Miro1's ability to regulate mitophagy is not intrinsically linked to its role in regulating kinesin-mediated anterograde mitochondrial transport as the expression of the mitophagy defective mutant, Miro1$^{5R}$, completely rescues mitochondrial redistribution in Miro$^{DKO}$ cells (Appendix Fig S4A–C).

Interestingly, we also observed that the expression of Miro1$^{allR}$ delayed the damage-induced translocation of Parkin to a greater extent than the absence of all Miro, suggesting that Miro1 not only acts as a receptor for Parkin translocation but may interfere with the normal progression of mitophagy by additional means when not ubiquitinated, effectively blocking Parkin translocation (Fig 3A–C). It has been suggested that the rapid ubiquitination of Mfn2 and consequent p97-dependent extraction from the OMM could act as a gating mechanism to allow widespread Parkin ubiquitination of other mitochondrial substrates and facilitate the progression of mitophagy linking the disassembly of the ER and mitochondria interactions to mitophagy progression (McLelland & Fon, 2018). Miro1 has also been shown to be an important regulator of ER–mitochondria contact sites (ERMCS) (Modi *et al*, 2019) and therefore results in a good candidate in being involved in such a gating mechanism. Indeed, we have shown that blocking Miro1 ubiquitination protected Miro1 from degradation in $^{Flag}$Parkin-expressing SH-SY5Y cells, a model system that triggers a very robust mitophagy response after mitochondrial damage induction (Fig 2D, F and G). To more accurately characterise Miro1 ubiquitination-dependent degradation,

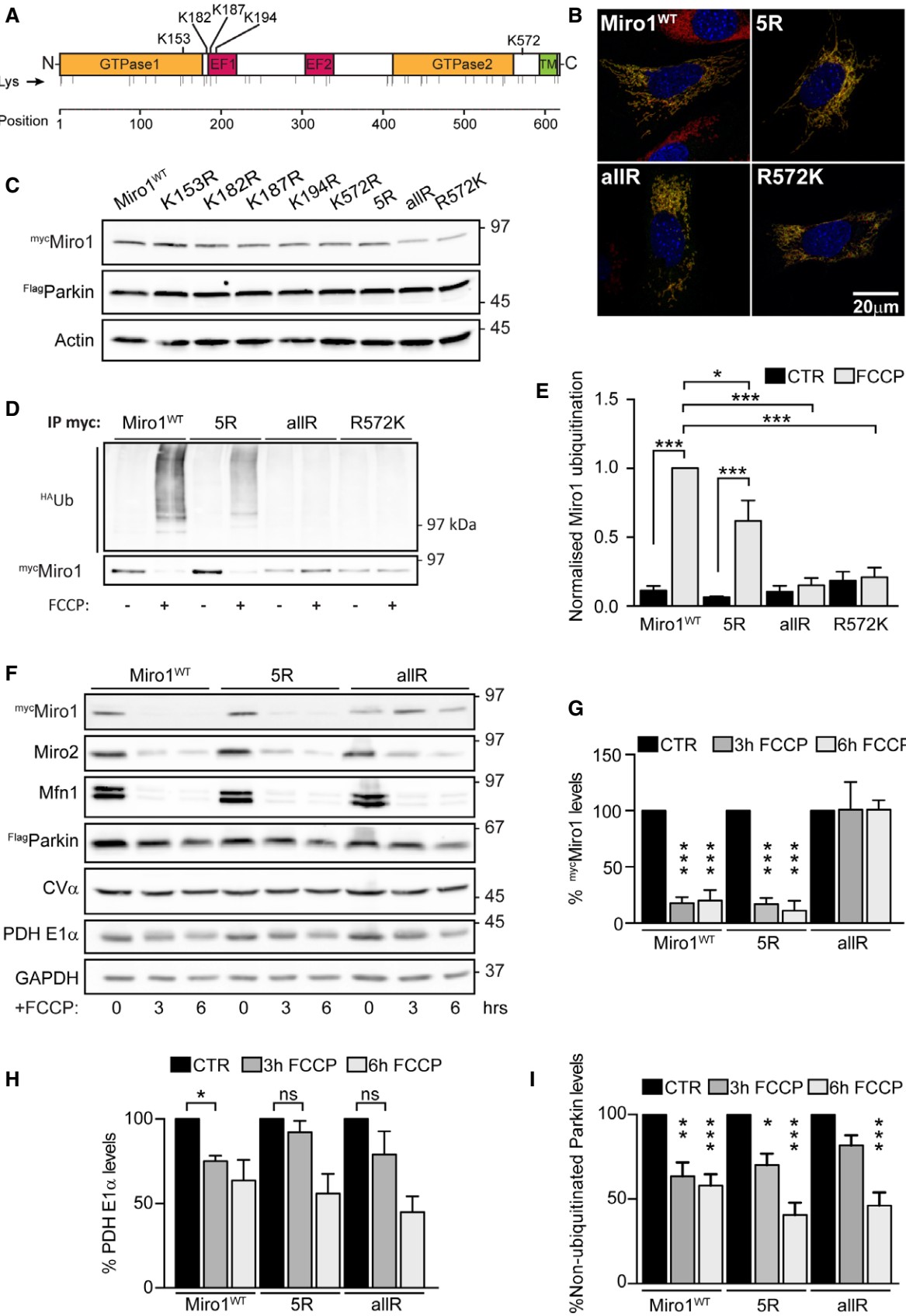

Figure 2.

◀

**Figure 2. Miro1 is promiscuously ubiquitinated during the mitophagic process.**

A Schematic representation of hMiro1 sequence highlighting domains and lysine residues (Lys) in its sequence. Key lysine residues reported to be ubiquitinated upon mitochondria damage and mutated to generate the $^{myc}$Miro1$^{5R}$ mutant construct are highlighted.

B Representative images showing full-length control and selected Miro1 lysine mutants (Miro1$^{5R}$, Miro1$^{allR}$ and Miro1$^{R572K}$) expressing in Miro$^{DKO}$ MEF cells. All constructs (myc-tag, green) localise to mitochondria (Tom20, red).

C Immunoblot showing efficient expression of Miro1 lysine mutants in $^{Flag}$Parkin-expressing SH-SY5Y cells.

D, E Ubiquitination assay showing that Miro1 lysine mutants have reduced ubiquitination upon FCCP treatment (1 h, 10 μM) in $^{Flag}$Parkin-overexpressing SH-SY5Y. The effect is quantified in (E) ($n = 4$ independent experiments for Miro1$^{WT}$ and Miro1$^{5R}$ and $n = 3$ for Miro1$^{allR}$ and Miro1$^{R572K}$, ANOVA with Sidak's *post hoc* test).

F–I Representative western blot showing a degradation assay in $^{Flag}$Parkin-overexpressing SH-SY5Y cells transfected with Miro1$^{WT}$, Miro1$^{5R}$ or Miro1$^{allR}$ constructs and treated with FCCP (10 μM) for 3 or 6 h. Quantification of Miro1 levels (G), the matrix protein PDH-E1α in (H) and Parkin levels (I) ($n = 4$ independent experiments; ANOVA with Sidak's *post hoc* test).

Data information: Error bars represent SEM. Significance: *$P < 0.05$, **$P < 0.01$ and ***$P < 0.001$.
Source data are available online for this figure.

we repeated these experiments in our Miro$^{DKO}$ cells, avoiding any interference from endogenous Miro1. We also included shorter time points after mitochondrial damage induction to measure rapid changes in Miro1 stability. Again, we observed that while Miro1$^{WT}$ is rapidly degraded after mitochondrial depolarisation, blocking Miro1 ubiquitination with the Miro1$^{allR}$ construct completely protected Miro1 from degradation (Fig 3D). Interestingly, in this setting we observed that the Miro1$^{5R}$ version, which is only partially ubiquitinated under damage, was also partially protected from degradation when compared to Miro1$^{WT}$ (Fig 3D) indicating that ubiquitination levels in Miro1 after mitochondrial damage correlate with its degradation rate.

To test whether the observed delay in Parkin recruitment and impairment of Miro1 degradation in cells expressing the ubiquitination mutants translates into a reduced efficiency of mitochondrial clearance, we quantified the fraction of cells without any Tom20 signal in mitochondria after 24 h of FCCP treatment (which has been previously used as an indicator of mitochondrial clearance (Narendra *et al*, 2008, 2010b)) in cells re-expressing Miro1 constructs. We observed a dramatic reduction of damage-induced mitochondrial turnover in both ubiquitination mutants, Miro1$^{5R}$- and Miro1$^{allR}$-expressing cells (Fig 3E and F). Interestingly, we often observed high expression levels of Miro1$^{5R}$ and Miro1$^{allR}$, but not Miro1$^{WT}$, even at 24 h after FCCP treatment (Fig 3E) supporting that ubiquitination defects result in the protection of Miro1 from degradation and in the disruption of the mitophagic process.

Altogether, these results indicate that in MEFs, blocking the ubiquitination and subsequent degradation of Miro1 in depolarised mitochondria may act in concert to interfere with the normal recruitment of Parkin and with mitophagy progression. This is in agreement with recent studies that have described impaired Miro1 degradation under mitochondrial damage in PD patients' fibroblasts (Hsieh *et al*, 2016) suggesting that failing to degrade Miro1 after mitochondrial damage may impair the normal progression of mitophagy. Thus, both the ubiquitination and the degradation of Miro1 appear to be important in ensuring the efficient stabilisation of Parkin on the mitochondrial membrane and the subsequent clearance of damaged mitochondria.

## Miro1 is critically important for recruiting Parkin to mitochondria to trigger mitophagy in neurons

Although Miro$^{DKO}$ MEFs provide a powerful system to study the Miro dependency of mitophagy, neurons represent a more

physiologically and pathologically relevant model. There is controversy with respect to the optimal conditions and timescales that enable Parkin translocation without significant cytotoxicity (Van Laar *et al*, 2011; Wang *et al*, 2011b; Cai *et al*, 2012; Joselin *et al*, 2012) and without the need of using caspase inhibitors (Cai *et al*, 2012; Lazarou *et al*, 2015) which can potentially interfere with the very process in study (Wang *et al*, 2011a). Therefore, we initially investigated conditions favourable for driving Parkin translocation in $^{YFP}$Parkin-transfected mouse primary cultured neurons, without leading to significant cell death. Treatment with commonly used concentrations of FCCP or antimycin-A led to a significant amount of cell death within 5 h, close to the levels of death induced by an excitotoxic glutamate insult (Appendix Fig S5A and B). In contrast, treatment with 1 μM valinomycin (Val), a potassium ionophore that depolarises mitochondria without affecting the pH gradient (Puschmann *et al*, 2017), led to a rapid remodelling of the mitochondrial network, followed by Parkin translocation in both somas and neurites (Fig 4A and Appendix Fig S6) without sustaining significant neuronal death (Appendix Fig S5A and B). Parkin initially accumulated in hot spots on the mitochondria, which eventually became Parkin rings entirely surrounding mitochondria (Fig 4A), as could be clearly observed in high-resolution (Airyscan) imaging, a method that increases imaging resolution by a factor of 1.7 in comparison with conventional confocal microscopy (Fig 4B). In stark contrast to WT neurons, we observed a dramatic delay in Parkin translocation in Miro1$^{KO}$ neurons (Fig 4A and Appendix Fig S6), as could be observed by reduced overlap between Parkin and the mitochondrial marker MtDsRed in line scans and in the extent of co-localisation between $^{YFP}$Parkin and MtDsRed (Fig 4C and D). Subsequent to the valinomycin-induced mitochondrial remodelling that occurs during the first 3 h of treatment, we could also observe a loss of somatic mitochondrial content as mitochondria begin to be cleared by the mitophagic process, which was not observed in Miro1$^{KO}$ neurons (Fig 4E and F).

As PINK1 is the only known kinase that phosphorylates ubiquitin, antibodies specific to S65-phospho-ubiquitin (pS65-Ub) can be used as a measure of PINK1-dependent mitophagy (Kane *et al*, 2014; Koyano *et al*, 2014; Fiesel *et al*, 2015) without the need for exogenous expression of fluorescently tagged Parkin. In WT neurons, pS65-Ub appeared in somatodendritic clusters following 1 μM valinomycin treatment and increased in number with longer treatment time, while the appearance of pS65-Ub puncta in Miro1$^{KO}$ neurons was severely delayed (Fig 4G and H). The specificity of the pS65-Ub signal in WT and Miro1$^{KO}$ neurons was confirmed by the absence of

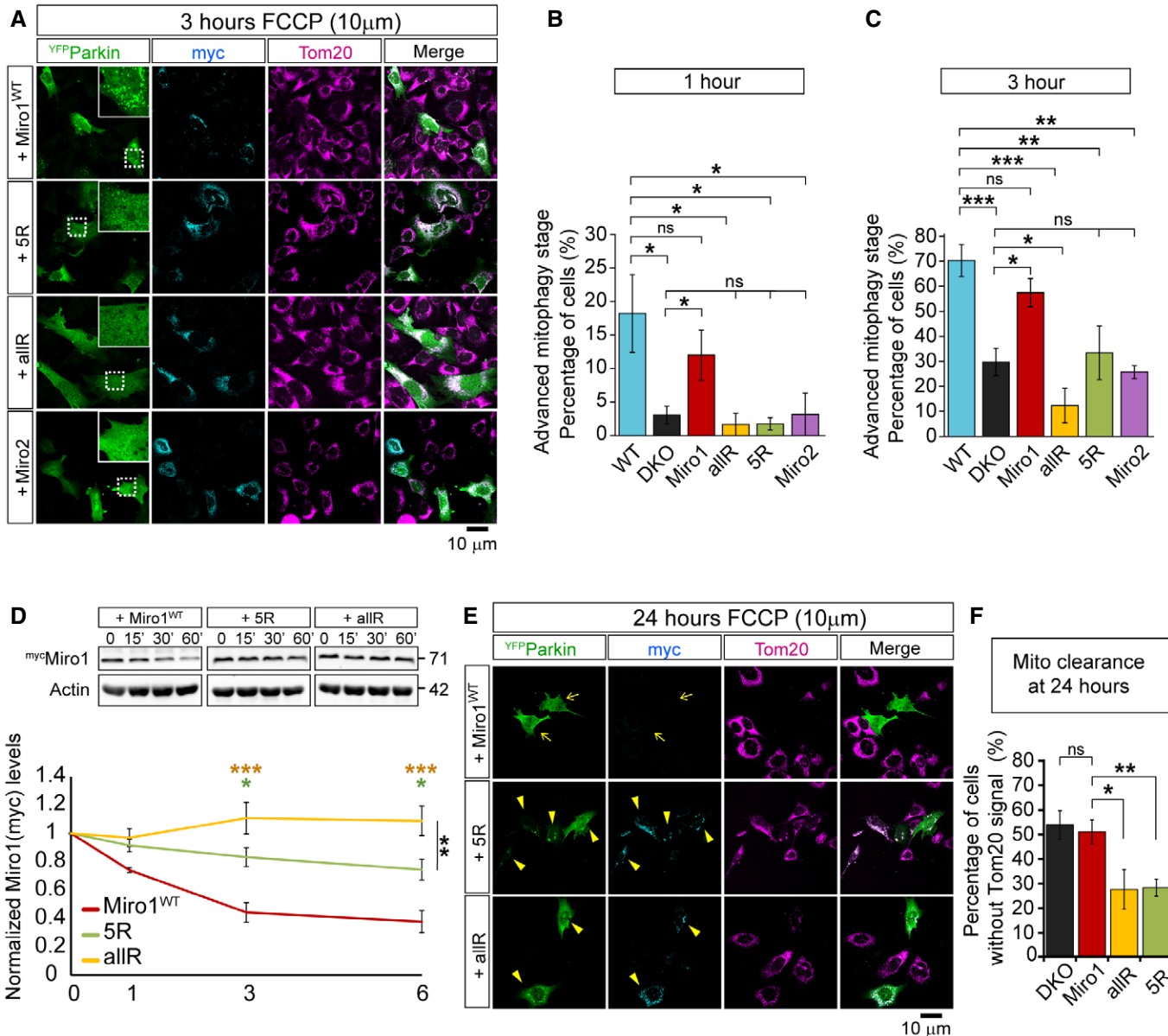

**Figure 3. Miro1 ubiquitination and degradation are required for mitochondrial clearance through mitophagy.**

A   Representative images at 3 h of FCCP treatment (10 μM) of YFPParkin-expressing (green) MiroDKO cells co-transfected with the specified myc-tagged versions of Miro1 (cyan). Tom20 (red) was used to reveal the mitochondrial network. A detail of YFPParkin translocation onto the mitochondrial network is also shown for each condition.

B, C   Quantification of the fraction of cells showing advanced stages of the mitophagic process (cells showing mitochondrial aggregation due to advanced Parkin translocation or the complete translocation of Parkin onto all the mitochondrial network) at 1 h (B) or 3 h (C) after FCCP treatment (data collected from at least 3 independent experiments: WT *n* = 7, MiroDKO *n* = 8, +mycMiro1WT = 7, mycMiro15R = 3, mycMiro1allR = 3; mycMiro2 = 3; one-way ANOVA with Dunnett post-test).

D   Western blots and quantification of Miro1 degradation in MiroDKO cells co-expressing YFPParkin and the indicated myc-tagged Miro1 constructs and treated with FCCP (10 μM) for the indicated times (*n* = 3 different experiments; two-way ANOVA with Dunnett post-test; comparisons between Miro1WT and Miro1allR are represented with yellow asterisks, comparisons between Miro1WT and Miro15R are represented in green, and comparisons between Miro15R and Miro1allR are represented in black).

E   Representative images at 24 h of FCCP treatment (10 μM) of YFPParkin-expressing (green) MiroDKO cells co-transfected with the specified myc-tagged versions of Miro1 (cyan). Tom20 (red) was used to reveal the mitochondrial network. Expression of both ubiquitin mutants (Miro15R and Miro1allR – arrowheads) protects from the mitophagic clearance of mitochondria (arrows) after damage.

F   Quantification of the fraction of cells showing the complete loss of Tom20 signal of mitochondria after 24 h of FCCP treatment (data collected from at least three independent experiments: MiroDKO *n* = 8, mycMiro1WT = 7, mycMiro15R = 3, mycMiro1allR = 3; one-way ANOVA with Dunnett post-test).

Data information: Error bars represent SEM. Significance: *$P < 0.05$, **$P < 0.01$ and ***$P < 0.001$.
Source data are available online for this figure.

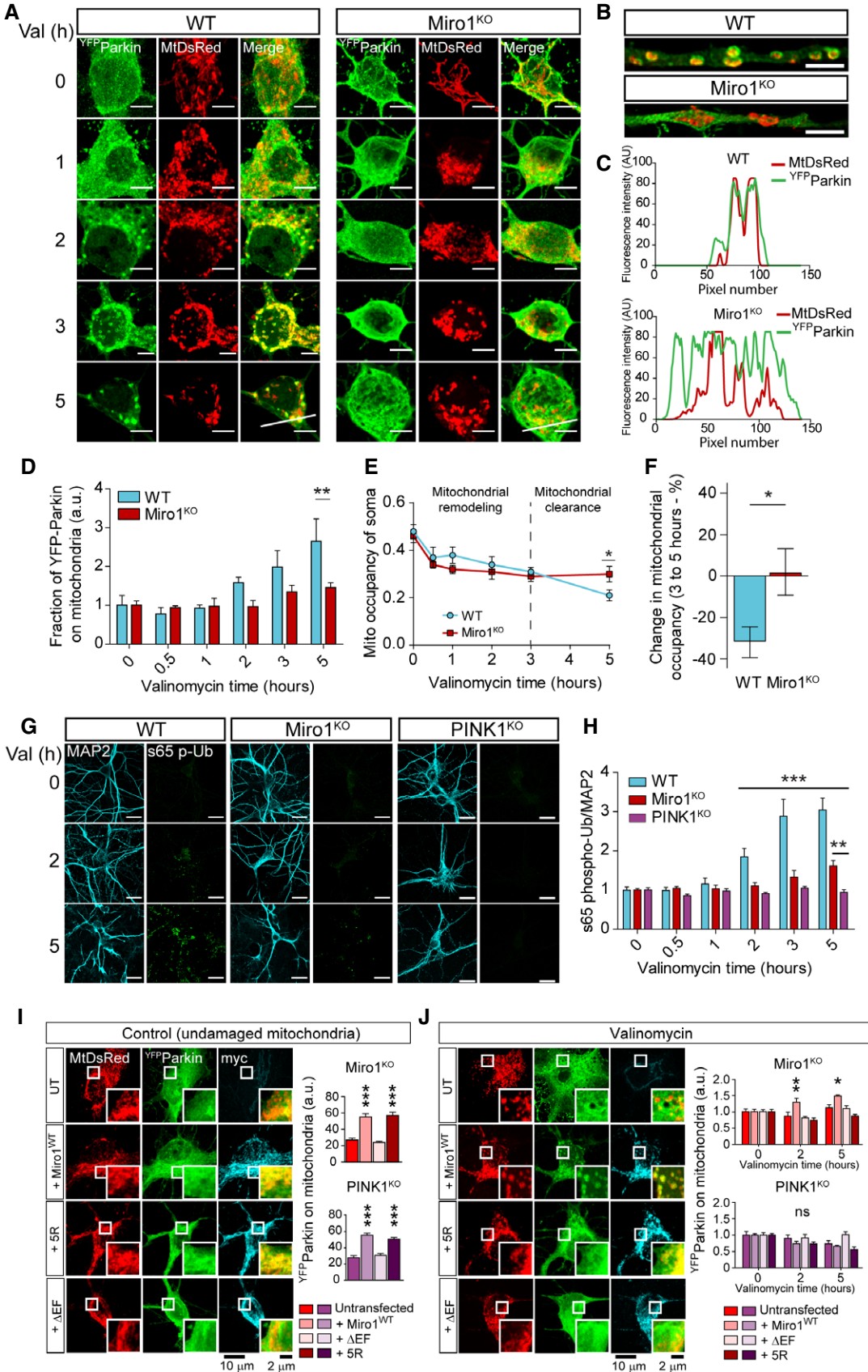

**Figure 4.**

Figure 4.   Miro1 expression and ubiquitination are required for Parkin recruitment to damaged mitochondria in primary neurons.

A   Representative confocal images of the soma of WT and Miro1[KO] cortical neurons expressing [YFP]Parkin and MtDsRed after valinomycin treatment at the indicated time points (scale bars = 5 μm).
B   Airyscan confocal images of Parkin recruitment after 5 h of valinomycin treatment in WT and Miro1[KO] neuronal processes (scale bars = 2.5 μm).
C   Fluorescent linescans of [YFP]Parkin and MtDsRed signal in WT and Miro1[KO] neurons after 5 h of valinomycin treatment (lines shown in A).
D   Quantification of Parkin recruitment to mitochondria ([YFP]Parkin signal overlapping MtDsRed signal. Intensity adjusted and normalised to $t = 0$) in WT and Miro1[KO] neurons following valinomycin treatment ($n = 15$ cells for all conditions per genotype over 3 neuronal preparations; two-way ANOVA).
E   Quantification of mitochondrial occupancy (area of MtDsRed signal in soma/entire area of soma) in the somas of WT and Miro1[KO] neurons following valinomycin treatment ($n = 15$ cells for all conditions per genotype over 3 neuronal preparations; two-way ANOVA).
F   Quantification of mitochondrial clearance in somas of WT and Miro1[KO] neurons between 2 and 5 h of valinomycin treatment ($n = 15$ cells for all conditions per genotype over 3 neuronal preparations; unpaired *t*-test).
G   Representative confocal images of WT, Miro1[KO] and PINK1[KO] cortical neurons immunostained with MAP2 (cyan) and pS65-Ub (green) after valinomycin treatment (scale bars = 20 μm).
H   Quantification of pS65-Ub signal intensity within MAP2 signal (normalised to MAP2 area and $t = 0$) in Miro1[WT], Miro1[KO] and PINK1[KO] neurons following valinomycin treatment ($n = 3$, 4 and 3 embryos from WT, Miro1[KO] and PINK1[KO] embryos, respectively, 6 ROIs per condition; two-way ANOVA).
I   Representative confocal images of the soma of Miro1[KO] cortical neurons expressing Miro1[WT] and mutant forms of Miro1, [YFP]Parkin and MtDsRed without valinomycin treatment (scale bars = 10 μm). Quantification of Parkin co-localisation with MtDsRed ([YFP]Parkin signal overlapping MtDsRed signal. Intensity adjusted) in the soma of Miro1[KO] and PINK1[KO] cortical neurons ($n = 12$ cells for all conditions per genotype over 3 neuronal preparations; one-way ANOVA).
J   Representative confocal images of the soma of Miro1[KO] cortical neurons expressing WT and mutant forms of Miro1, [YFP]Parkin and MtDsRed after 5 h of valinomycin treatment (scale bars = 10 μm). Quantification of Parkin recruitment to mitochondria (normalised to $t = 0$) in Miro1[KO] and PINK1[KO] cortical neurons following valinomycin treatment ($n = 12$ cells for all conditions per genotype over 3 neuronal preparations, two-way ANOVA).

Data information: Error bars represent SEM. Significance: *$P < 0.05$, **$P < 0.01$ and ***$P < 0.001$.

immunofluorescence in neurons derived from PINK1[KO] embryos (Fig 4G and H). Thus, also in neurons, Miro1 appears to be critically important for the stabilisation of Parkin on the OMM and for the formation of pS65-Ub chains upon mitochondrial damage, which are critical steps for damage-induced mitochondrial clearance.

## Ubiquitination of Miro1 recruits Parkin to polarised mitochondria and during damage-induced mitophagy in neurons

To further investigate the mechanisms underlying the role of Miro1 in the mitophagic process in neurons, we performed rescue experiments in Miro1[KO] neurons by expressing WT or mutant forms of Miro1. Intriguingly, when performing untreated control experiments, we observed that exogenous expression of Miro1[WT] in Miro1[KO] neurons induced Parkin recruitment to mitochondria even in the absence of valinomycin treatment (Fig 4I). This recruitment of Parkin to undamaged mitochondria was recently shown to be dependent on the ability of Miro1 to sense calcium in non-neuronal cells (Safiulina *et al*, 2019). In these experiments, Parkin was enriched onto tubular, elongated and reticular mitochondria unlike the rings of Parkin surrounding fragmented mitochondria after valinomycin-induced damage (Fig 4I). Similar to Miro1[WT], expression of Miro1[5R] also induced enrichment of Parkin onto undamaged mitochondria, suggesting that the ubiquitination defect introduced in Miro1[5R] does not block the ability of Miro1 to bind Parkin in undamaged mitochondria (Fig 4I). We also included a Miro1 construct with mutations (E208K and E328K) in both EF-hand motifs (Miro1[ΔEF]), which abolish $Ca^{2+}$ binding and render Miro1 insensitive to calcium (Fransson *et al*, 2006) and that was recently shown to be poorly ubiquitinated and unable to recruit Parkin to mitochondria (Safiulina *et al*, 2019). As expected, the expression of Miro1[ΔEF] did not induce the translocation of Parkin to undamaged, polarised mitochondria in neurons (Fig 4I). Importantly, Parkin recruitment to polarised mitochondria was not dependent on PINK1 as both Miro1[WT] and Miro1[5R] were still able to recruit Parkin to undamaged mitochondria in PINK1[KO] hippocampal neurons (Fig 4I

and Appendix Fig S7A), indicating that Parkin recruitment by over-expression of Miro1 is independent of mitophagy activation. However, during valinomycin treatment, only expression of Miro1[WT], but not Miro1[5R], in Miro1[KO] neurons significantly increased Parkin translocation to mitochondria after 2 and 5 h (Fig 4J). In this case, such translocation was occluded in PINK1[KO] neurons indicating that under mitochondrial damage, Parkin recruitment by Miro1 is dependent on the activation of mitophagy and requires Miro1 to be ubiquitinated (Fig 4J and Appendix Fig S7B). Thus, Miro1 can bind and enrich Parkin on mitochondria in the absence of mitochondrial damage and independently of the activation of the mitophagic process by a mechanism that is dependent on calcium, while during mitophagy activation, the ubiquitination of Miro1 is paramount for the enrichment of Parkin onto damaged mitochondria. Therefore, the presence of Miro1 and its correct ubiquitination appear to be equally important for ensuring the recruitment and stabilisation of Parkin on the mitochondrial membrane during mitochondrial damage in neurons.

## Deletion of Miro1 leads to an upregulation of mitofusins and the mitophagic machinery *in vivo*

Defects in mitophagy can lead to an age-dependent accumulation of mitochondrial dysfunction underpinning neurological disease. To investigate whether altered mitophagy induced by the loss of Miro1 affects mitochondrial homeostasis under more physiological conditions, we investigated the expression levels of key mitophagy substrates in mouse brain, where neurons can develop on an extended timescale compared to cell cultures. Loss of Miro1 is lethal perinatally (Nguyen *et al*, 2014; Lopez-Domenech *et al*, 2016). To bypass the requirement of Miro1 for normal development and study the impact of Miro1 deletion in mitochondrial homeostasis in mature excitatory neurons, we used a conditional Miro1 knockout model crossed with a mouse line expressing the Cre recombinase under the control of the CaMKIIα promoter (Miro1[CKO] from hereon) (Mantamadiotis *et al*, 2002; Lopez-Domenech *et al*, 2016). In

addition, we analysed the impact of knocking out Miro2, which can also be a substrate for damage-induced ubiquitination (Mertins *et al*, 2013; Sarraf *et al*, 2013; Udeshi *et al*, 2013). Hippocampal lysates were prepared from 4 and 12 month-old WT, Miro1$^{CKO}$ and Miro2$^{KO}$ animals and studied by western blotting. Loss of Miro1 or Miro2 had no effect on basal mitochondrial content at either 4 or 12 months, as observed by no changes in ATP5α (the alpha subunit of the ATPase in complex V of the electron transport chain) (Fig 5A and B). At 4 months of age, no difference in Parkin and PINK1 levels was observed between WT, Miro1$^{CKO}$ and Miro2$^{KO}$ animals; however, our western blot analysis at 12 months of age revealed that there was a significant increase in PINK1 levels and the appearance of an upper band of Parkin, suspected of being an auto-ubiquitinated form (Chaugule *et al*, 2011; Wauer & Komander, 2013) (Fig 5A and B). This indicates that Miro1$^{CKO}$ neurons might sustain an over-activation of the mitophagic pathway in an effort to counteract accumulated mitochondrial damage. To test for variations in the activity of Parkin-mediated mitophagy *in vivo,* the levels of Mfn1, Mfn2 and VDAC1 – well-characterised Parkin substrates – were also probed (Gegg *et al*, 2010; Geisler *et al*, 2010; Chen & Dorn, 2013; Sarraf *et al*, 2013). Interestingly, at 4 months, an increase in Mfn1 was observed in Miro1$^{CKO}$ in comparison with WT animals, though no increase in VDAC1 was observed (Fig 5A and B). Strikingly, the levels of both Mfn1 and Mfn2 in the Miro1$^{CKO}$ at 12 months were substantially and consistently increased in comparison with WT hippocampal lysates, an effect that was not observed in the Miro2$^{KO}$ mice (Fig 5A and B). Importantly, the upregulation of Mfn1 and Mfn2 was not due to increased transcription (Appendix Fig S8A and B) suggesting that the increased protein levels respond to inefficient targeting by the PINK1/Parkin pathway. To further assess this, we performed an ubiquitination assay on Miro1$^{KO}$ neurons *in vitro*. We treated mature neurons (DIV14) cultured from Miro1$^{KO}$ embryos and litter matched WT controls with MG-132, a proteasomal inhibitor, for 4 h to prevent clearance of ubiquitinated substrates via the proteasome. Immunoprecipitation of Mfn2 followed by western blotting for ubiquitin revealed that Mfn2 is significantly less ubiquitinated in Miro1$^{KO}$ neurons compared to WT controls (Appendix Fig S8C and D). In summary, conditional Miro1 deletion in mouse brain, *in vivo*, leads to an upregulation of the mitophagic machinery in an age-dependent manner.

## Loss of Miro1 in principal neurons induces the appearance of megamitochondria with altered morphology and ultrastructure

The dramatic upregulation of Mfn1 and Mfn2 in Miro1$^{CKO}$ principal neurons suggests a pathological reconfiguration of the mitochondrial network due to an altered balance of the fission/fusion dynamics caused by disruption of PINK1/Parkin-mediated mitophagy. To further address this *in vivo,* we crossed the Miro1$^{CKO}$ mouse with a mouse line that allows conditional Cre recombinase-dependent expression of the mitochondrial reporter mitoDendra (Appendix Fig S9A). Crossing the mitoDendra line with Cre recombinase led to robust expression of mitoDendra in CamKIIα-driven Cre expressing cells. This approach allowed us to visualise the mitochondrial network, specifically in principal neurons of cortex and hippocampus from aged animals (Fig 5C). Using a similar approach, we generated age-matched Miro2$^{KO}$ mice with mitochondria labelled with mitoDendra in CamKIIα-expressing cells which may help in

informing about specificities and similarities between Miro1 and Miro2 in regulating mitochondrial homeostasis *in vivo*. In control mice (heterozygous for the Miro1 conditional allele – Miro1(Δ/+)), mitoDendra-labelled mitochondria appeared as an elongated network of various sized mitochondrial elements very similar to the mitoDendra signal observed in Miro2$^{KO}$ animals (Fig 5D). In stark contrast, mitoDendra-labelled mitochondria in Miro1$^{CKO}$ cells at 1 year exhibited a dramatic remodelling of somatic mitochondria revealing large mitochondrial units that were absent in cell bodies of control or Miro2$^{KO}$ neurons (Fig 5D and Appendix S9B). To address the impact of deleting Miro1 on the mitochondrial network with higher resolution, we performed electron microscopy analysis from control or Miro1$^{CKO}$ tissues. While mitochondria in control neurons from 4-month-old animals appeared with the expected elongated morphology and with the usual ultrastructure of membrane cristae filling the intra-mitochondrial space (Fig 5E), we observed that mitochondria in Miro1$^{CKO}$ cells appeared swollen and rounded and the intra-mitochondrial space less electron dense than that from control or Miro2$^{KO}$ cells (Fig 5E). Importantly, this effect was greatly exacerbated in neurons from 12-month-old animals (Fig 5E) indicating that alterations in mitochondrial morphology due to loss of Miro1 are progressive. The mitochondrial phenotype induced by Miro1 deletion is similar to previous reports of accumulated and hyperfused "giant" or megamitochondria identified in a number of models where either mitochondrial fission/fusion proteins or mitophagy has been altered (Chen *et al*, 2007; Kageyama *et al*, 2012; Kageyama *et al*, 2014; El Fissi *et al*, 2018; Yamada *et al*, 2018a; Yamada *et al*, 2018b). Immunostaining of coronal brain slices from 12-month-old animals with specific antibodies showed a large increase in both Mfn1 and Mfn2 levels specifically in the mitochondria of principal neurons from the Miro1$^{CKO}$ brains when compared to WT neurons (Fig 5F and G) allowing us to confirm that the increase in Mfn1 and Mfn2 observed in brain lysates was specifically happening in the neurons where Miro1 was deleted and was not due to a dysregulation of tissue homeostasis. Interestingly, we observed that a proportion of Miro1$^{CKO}$ cells presenting giant mitochondria showed an accumulation of ubiquitin surrounding the aberrant mitochondrial particles similar to what happens during ageing or Lewy body disease (Appendix Fig S9C) (Hou *et al*, 2018). Altogether, the severe mitochondrial remodelling and disruption of mitochondrial ultrastructure observed in Miro1$^{CKO}$ neurons *in vivo* suggests that mitochondrial homeostasis might be severely compromised by the loss of Miro1 in mature neurons.

## Long-term loss of Miro1 *in vivo* leads to the activation of the integrated stress response

Animal models in which either mitochondrial fission and fusion or mitophagy has been altered, have been shown to induce the ISR, a protective pathway that leads to a reduction in global protein synthesis rates. However, sustained activation of this pathway leading to a chronic reduction in the translation of vital proteins can result in neuronal death (Munoz *et al*, 2013; Celardo *et al*, 2016; Restelli *et al*, 2018). Activation of the ISR is mediated by the activity of 4 discrete kinases (PERK, HRI, PKR and GCN2) which all converge onto one phosphorylation site at serine 51 of the eukaryotic initiation factor 2 alpha (eIF2α). Therefore, antibodies specific to pS51-eIF2α can be used as a measure of activation of the ISR. Strikingly, the levels of

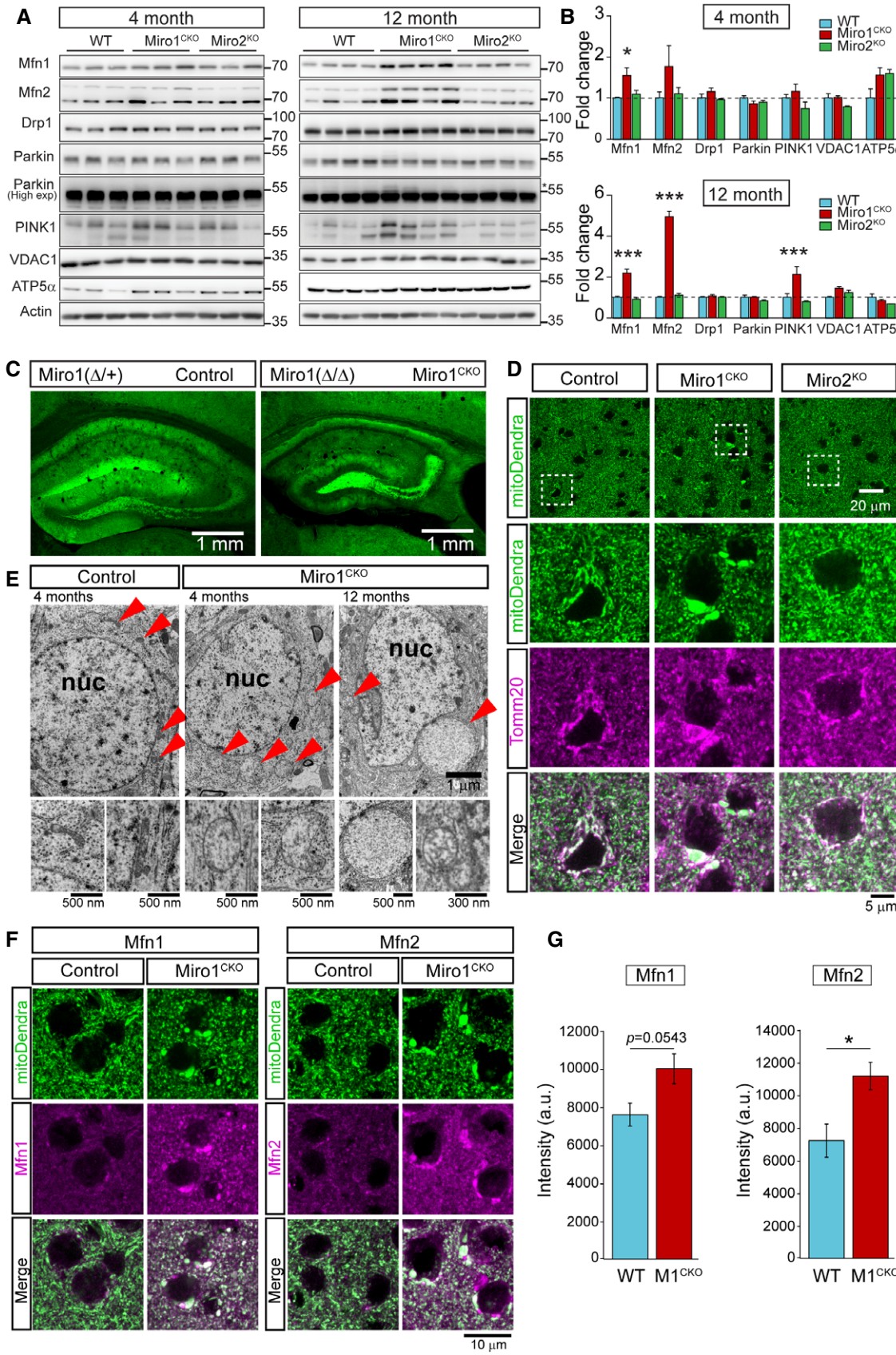

**Figure 5.**

◄

**Figure 5.   Loss of Miro1 in mature neurons associates with age-dependent accumulation of defective mitochondria in neurons and an increase in mitophagy machinery.**

A      Representative blots of 4- and 12-month-old hippocampal lysates from control, Miro1[CKO] and Miro2[KO].
B      Quantification of protein levels in 4-month lysates and 12-month lysates (*n* = 3 animals/genotype at 4 months and *n* = 4 animals/genotype at 12 months; one-way ANOVA with *post hoc* Dunnett's test).
C      Representative images of hippocampal regions from control and Miro1[CKO] crossed with mitoDendra animals.
D      Details from cortical regions of control, Miro1[CKO] and Miro2[KO] animals at 12 months of age showing large mitochondrial structures occurring only in the case of Miro1[CKO] animals.
E–G   Electron microscopy images of control and Miro1[CKO] brains showing the ultrastructure of the mitochondrial compartment. The mitochondrial units in Miro1[CKO] cells are enlarged and show altered cristae structure at 4 months of age. This process is progressive as mitochondrial structures appear larger and with reduced intra-mitochondrial complexity at 12 months of age. Representative images (F) and quantification (G) of Mfn1 and Mfn2 staining in cortical slices from 12-month WT and Miro1[CKO] mice (*n* = 4 animals/genotype; Student's *t*-test).

Data information: Error bars represent SEM. Significance: *$P < 0.05$, **$P < 0.01$ and ***$P < 0.001$.
Source data are available online for this figure.

pS51-eIF2α in 12-month-old Miro1[CKO] hippocampal lysates (age which corresponds to the largest increase in Mfn1/2) were substantially and consistently increased in comparison with WT hippocampal lysates, an effect that was not observed in the Miro2[KO] mice (Fig 6A and B). Total levels of eIF2α were unchanged between all 3 genotypes allowing us to conclude the increase of pS51-eIF2α seen in Miro1[CKO] brains is due to an increase in phosphorylation of eIF2α. To confirm this, coronal brain slices from 12-month-old control and Miro1[CKO] mitoDendra-expressing mice were stained with pS51-eIF2α. In agreement with the western blotting data, there was a large increase in pS51-eIF2α staining in the soma of MAP2+ neurons in the cortex of Miro1[CKO] mice compared with control mice (Fig 6C and D). In order to establish whether the activation of the ISR in Miro1[CKO] brains correlates with the formation of megamitochondria, we established two populations of neurons within Miro1[CKO] brains: those with megamitochondria and those without. Using mitoDendra signal, megamitochondria were defined as mitochondria with an area > 2.5 µm², consistent with previous reports (Yamada *et al*, 2018b). Strikingly, there was a significant increase in pS51-eIF2α staining in neuronal soma containing megamitochondria compared to those without (Fig 6E–G). Collectively, our data indicate that long-term disruption of mitochondrial homeostasis *in vivo* by Miro1 deletion leads to increased levels of mitofusins, formation of megamitochondria and ISR activation.

## Discussion

Here, we characterise the critical role of Miro1 ubiquitination and degradation for the initiation and progression of PINK1/Parkin-dependent mitophagy using Miro1/2 DKO mouse embryonic fibroblasts and primary Miro1 and PINK1 constitutive knockout mouse neurons. In addition, using a conditional Miro1 knockout mouse line with Miro1 deletion in principal neurons of the hippocampus and cortex, we show that long-term disruption of mitochondrial homeostasis *in vivo* leads to increased levels of Mfn1/2, remodelling of the mitochondrial network and induction of the integrated stress response.

We, and others, have previously shown that Miro1 promotes the recruitment and stabilisation of Parkin onto damaged mitochondria (Birsa *et al*, 2014; Shlevkov *et al*, 2016; Safiulina *et al*, 2019). This would suggest that Parkin binding by Miro, in conjunction with Parkin activation by phospho-ubiquitin, could act together to

stabilise Parkin on the OMM, providing a mechanism to tune Parkin levels to that of substrates on the OMM. In agreement with a key role for Miro in stabilising mitochondrial Parkin, we now show that deleting both Miro proteins in Miro[DKO] cells leads to dramatically reduced Parkin translocation and accumulation onto mitochondria upon damage. Thus, our results further support our previous proposal that Miro forms part of a Parkin receptor complex on the OMM important for tuning Parkin-mediated mitochondrial quality control (Birsa *et al*, 2014). Intriguingly, however, we find that Miro ubiquitination also appears to be required to allow the later stages of mitophagy to progress. More specifically, we show that mutation of lysine residues within Miro1 reduces Parkin recruitment upon mitochondrial damage and delays autophagic clearance of mitochondria. It is possible that the slowed Parkin translocation observed upon rescue with Miro1[allR] and Miro1[5R] may be because Miro ubiquitination is specifically required for Parkin stabilisation and amplification of the ubiquitin signal. Indeed, we previously showed that the pool of ubiquitinated Miro remains localised on the OMM for some considerable time prior to it being turned over, dependent on the activity of the proteasome (Birsa *et al*, 2014). Thus, Miro ubiquitination (rather than ubiquitination-induced Miro degradation) may directly act as a rapid signal on the OMM for the mitophagic process.

A previous report analysing Miro ubiquitination by Parkin *in vitro* (Klosowiak *et al*, 2016) proposed an important role for Miro1 residue K572 in the Parkin-mediated damage response. In our experiments, we found that the Miro1 K572 ubiquitination site alone was insufficient for mediating Miro1 ubiquitination and turnover, even in dopaminergic SH-SY5Y neuroblastoma cells stably expressing Parkin, which show high levels of damage-induced Parkin activity. Although Miro1 K572 ubiquitination by Parkin may be important and could, for example, impact the rate or kinetics of Miro ubiquitination by Parkin, importantly, we did not find any substantial ubiquitination of Miro1[R572K] (where K572 is the only lysine replaced in the Miro1[allR]) when compared to Miro1[allR], which cannot be ubiquitinated. Thus, ubiquitination of Miro1 by Parkin requires the ubiquitination of multiple lysine residues to regulate Miro stability. This is consistent with a number of ubiquitination sites being identified within Miro1 upon damage (Povlsen *et al*, 2012; Sarraf *et al*, 2013; Kazlauskaite *et al*, 2014; Ordureau *et al*, 2014; Ordureau *et al*, 2015; Wu *et al*, 2015; Boeing *et al*, 2016) and at steady state (Kim *et al*, 2011; Wagner *et al*, 2011; Wagner *et al*, 2012; Mertins *et al*, 2013; Udeshi *et al*, 2013; Lumpkin *et al*, 2017). It is important to note that our current data do not preclude the

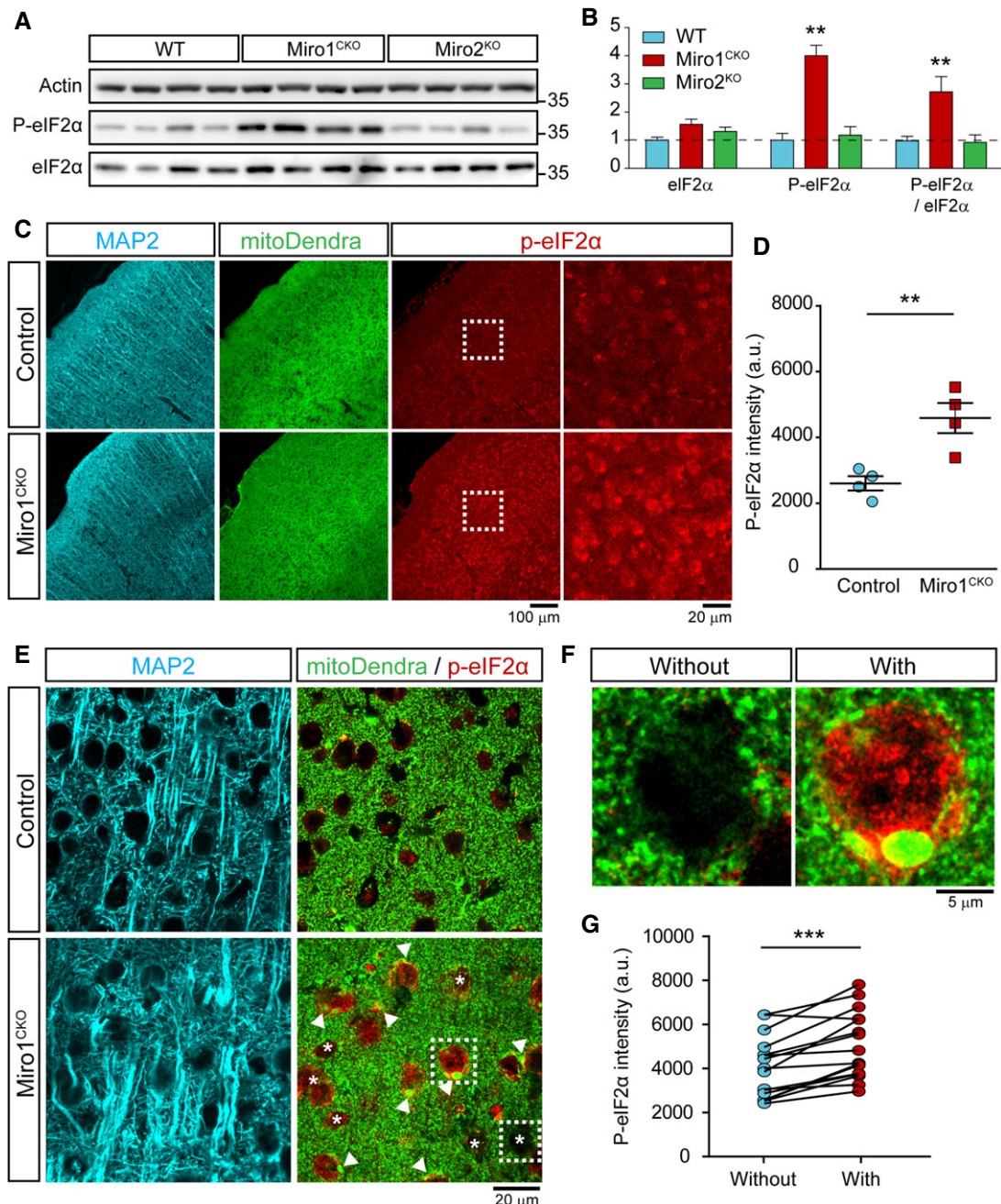

**Figure 6. Long-term loss of Miro1 in mature neurons leads to the activation of the integrated stress response.**

A   Representative western blot images of brain lysates from 12-month WT, Miro1[CKO] and Miro2[KO] mice immunoblotted with the antibodies stated.

B   Quantification of eIF2α and P-eIF2α band intensity (normalised to actin band intensity) from 12-month WT, Miro1[CKO] and Miro2[KO] brain lysates. Phosphorylation of eIF2α is defined as P-eIF2α/eIF2α ($n = 4$ mice per genotype, unpaired $t$-test).

C   Representative confocal images of cortical regions of 12-month mitoDendra-crossed control and Miro1[CKO] mice stained with MAP2 and P-eIF2α.

D   Quantification of P-eIF2α signal intensity from cortical regions of 12-month mitoDendra-crossed control and Miro1[CKO] mice ($n = 4$ mice per genotype; unpaired $t$-test).

E   Representative confocal images (63×) of cortical regions of 12-month mitoDendra-crossed control and Miro1[CKO] mice stained with MAP2 and P-eIF2α. Arrows indicate the presence of megamitochondria (> 2.5 μm²) and stars indicate neuronal somas without megamitochondria.

F   Example zoom images of neuronal somas with and without megamitochondria from cortical regions of 12-month mitoDendra-crossed Miro1[CKO] mice stained with P-eIF2α.

G   Quantification P-eIF2α signal intensity (a.u.) from neuronal somas with and without megamitochondria from cortical regions of 12-month mitoDendra-crossed Miro1[CKO] ($n = 16$ sections from 4 mice; paired $t$-test).

Data information: Error bars represent SEM. Significance: *$P < 0.05$, **$P < 0.01$ and ***$P < 0.001$.

Source data are available online for this figure.

possibility that ubiquitination of K572 acts to prime subsequent ubiquitination of other sites. In fact, a recent study showed that expression of Miro1[K572R] delays Parkin recruitment to damaged mitochondria (Safiulina *et al*, 2019). Thus, it is possible that Miro1 ubiquitination at K572 could occur in a cell-specific manner or have a cell-specific impact on Miro ubiquitination kinetics, depending on the expression levels of Parkin or of other regulatory factors such as Miro phosphorylation state (Shlevkov *et al*, 2016) and Miro protein kinases or phosphatases. However, our work shows that K572 of Miro1 alone is not sufficient for driving mitophagic progression.

Intriguingly, we show that the presence of less and non-ubiquitinatable forms of Miro1 restrain mitochondrial clearance even more so than its loss, further highlighting the importance of Miro1 degradation for mitophagy. It has been suggested that the rapid ubiquitination of certain Parkin substrates, like Mfn2, can facilitate the release of dysfunctional mitochondria from the ER in a p97-dependent manner which acts as a gating mechanism to allow widespread Parkin ubiquitination of other mitochondrial substrates and facilitate the progression of mitophagy (McLelland & Fon, 2018). In support of this view, expression of PTPIP51, an ER/mitochondrial anchoring protein, results in the suppression of Parkin-mediated mitophagy, highlighting the role of the ERMCS in the regulation of mitophagy progression (Puri *et al*, 2019). Recently, it has been shown that Miro1 is present at the ERMCS (Modi *et al*, 2019) and alongside Mfn2 is one of two mitochondrial targets of p97 (Ordureau *et al*, 2020). Thus, it could be that the ubiquitination and subsequent degradation of Miro1 is required to release damaged mitochondria from the ER to facilitate their clearance, a process that cannot occur when less ubiquitinatable, and thus stabilised, forms of Miro1 are present on mitochondria.

Defects in mitochondrial homeostasis are a key part of the aetiology of neurodegenerative diseases; for example, mutations in PINK1 and Parkin are associated with early onset familial Parkinson's disease. Therefore, while work in mammalian cell lines provides mechanistic insight into the mitophagic process, the use of *in vitro* neuronal cultures and *in vivo* models is essential to our understanding of neurodegeneration. Using constitutive and conditional mouse knockout strategies, we also demonstrate that Miro1 and its ubiquitination are important for mitophagy in neurons. Importantly, we established conditions using the mitochondrial uncoupler valinomycin to allow for robust Parkin translocation and accumulation of pS65-Ub without causing the significant neuronal death observed with other commonly used inducers of mitophagy such as FCCP or antimycin-A. In accordance with an earlier report, we found Parkin translocation onto mitochondria to be considerably slower than in non-neuronal cells (Cai *et al*, 2012). Similar to our Miro[DKO] MEF studies, Miro1[KO] neurons exhibited dramatically reduced Parkin translocation onto mitochondria compared to WT neurons and a considerable delay in the appearance of pS65-Ub clusters following damage. This also correlated with reduced mitochondrial clearance at later time points. Likewise, damaged-induced Parkin recruitment to mitochondria and progression of mitophagy in neurons was also dependent on the ubiquitination of Miro1. Interestingly, when performing the rescue experiments, we found that Parkin is recruited to undamaged mitochondria when Miro1 is exogenously expressed in Miro1[KO] neurons, consistent with previous findings in non-neuronal cells (Safiulina *et al*, 2019). As this Parkin recruitment was still observed in the absence of PINK1 (in PINK1[KO] neurons),

this is unlikely to result in degradation of mitochondria as Parkin's E3-ligase activity is dependent on its phosphorylation by PINK1. Further, Parkin can then be recruited upon damage-induced stabilisation of PINK1 on the OMM and ubiquitination of Miro1, thus facilitating mass ubiquitination of Parkin substrates and the clearance of neuronal mitochondria.

The late onset of many neurodegenerative diseases suggests that damage can accumulate with time because of long-term defects in cellular processes. Given the defects in mitophagy caused upon loss of Miro1, we studied the *in vivo* consequences of the loss of Miro1 in aged animals. Importantly, we identified hallmarks of altered neuronal mitochondrial quality control *in vivo*. For example, we observed a significant increase in PINK1 levels and the appearance of an upper band of Parkin, suspected of being an auto-ubiquitinated form of Parkin in 12-month-old Miro1[CKO] mice (Chaugule *et al*, 2011; Wauer & Komander, 2013). Parkin auto-ubiquitination leads to the activation of Parkin E3-ligase activity, with sustained auto-ubiquitination being proposed to be indicative of a pathological state of Parkin activation (Chaugule *et al*, 2011). Furthermore, overexpression of PINK1 is known to cause mitochondrial dysfunction (Yang *et al*, 2008). Alongside the changes in PINK1 and Parkin in 12-month-old Miro1[CKO] mice, we also revealed a dramatic increase in the expression levels of Mfn1 and Mfn2. In addition, we found that ubiquitination of Mfn2 is significantly decreased when Miro1 is knocked out in cortical neurons *in vitro*. This is likely a consequence of reduced Parkin E3-ligase activity in Miro1[KO] neurons due to delayed Parkin recruitment to mitochondria. Interestingly, alterations in Mfn1 and Mfn2 have been observed in several models of mitophagic dysfunction and neurodegeneration (Chen & Dorn, 2013; Gong *et al*, 2015; Wang *et al*, 2015; Rocha *et al*, 2018). Overexpression and reduced ubiquitination of Mfn2 repress mitophagy (McLelland & Fon, 2018), perhaps further exacerbating the defects in mitophagy in Miro1[CKO] animals. Therefore, we conclude that long-term loss of Miro1 leads to profound changes in the mitophagy and mitochondrial fusion machinery.

Upregulation of the mitochondrial fusion machinery and in particular gain-of-function mutations of Mfn2 associated with Charcot–Marie–Tooth-2 (CMT2) pathology lead to mitochondrial remodelling and the accumulation of hyperfused "giant mitochondria" in neuronal somas (Santel & Fuller, 2001; El Fissi *et al*, 2018). Interestingly, studying mitochondrial morphology in aged Miro1[CKO] neurons *in vivo* revealed a dramatic hyperfusion of the mitochondrial network and large mitochondrial structures located in the cell bodies. This is similar to previous reports of collapsed mitochondrial structures identified in a number of models where either mitochondrial fission/fusion proteins or mitophagy has been altered (Chen *et al*, 2007; Kageyama *et al*, 2012; Kageyama *et al*, 2014; El Fissi *et al*, 2018; Yamada *et al*, 2018a; Yamada *et al*, 2018b).

Mouse models with disrupted mitochondrial dynamics in neurons can present sustained levels of cellular stress that lead to the activation of the integrated stress response (ISR) (Restelli *et al*, 2018). Alterations in Mfns either directly, in Mfn2 ablated MEFs, or as a consequence of dysfunctional mitophagy, in PD mutant flies, are known to induce the ISR, a protective pathway that leads to a reduction in global protein synthesis rates upon detection of cellular stress (Munoz *et al*, 2013; Celardo *et al*, 2016). The ISR is implicated in the pathology of numerous neurodegenerative diseases where the chronic reduction in translation of vital proteins leads to neuronal

death (Halliday & Mallucci, 2015). Intriguingly, we found that the formation of hyperfused megamitochondria in Miro1$^{CKO}$ brains led to initiation of the ISR *in vivo,* providing further links between the loss of Miro and neurodegeneration. The induction of the ISR is mediated by the activity of four discrete kinases, PERK, HRI, PKR and GCN2, all of which converge onto a single phosphorylation site of eIF2α at serine 51 (Wek, 2018); however, it is currently unclear which of these kinases is responsible for the increase observed in Miro1$^{CKO}$ neurons. Recent studies have shown that HRI and GCN2 can act as sensors for mitochondrial stress (Fessler *et al*, 2020; Guo *et al*, 2020; Mick *et al*, 2020); however, the majority of ISR activation seen in models of neurodegenerative disease is dependent on PERK (Smith & Mallucci, 2016). Interestingly, this includes mutant PINK1 and Parkin *Drosophila* models of genetic Parkinson's disease (Celardo *et al*, 2016). In these models, PERK is activated via the upregulation of Mfn and formation of Mfn bridges between damaged mitochondria and the ER. Inhibiting PERK activity or knocking down Mfn prevented ISR activation and was neuroprotective, irrespective of the mitochondrial dysfunction that remained (Celardo *et al*, 2016). In Miro1$^{CKO}$ brains, we show that pS51-eIF2α expression is significantly higher in neurons containing megamitochondria compared to those without. Additionally, we observed that Mfn expression is enriched on megamitochondria. Therefore, it may be that the ISR is activated via a similar Mfn- and PERK-dependent mechanism in the brains of Miro1$^{CKO}$ mice and going forward it will be important to establish what impact blockade of this pathway has on the neurodegeneration we have previously described in this model (Lopez-Domenech *et al*, 2016). Notwithstanding, our findings further implicate Miro1 dysfunction in the pathogenesis of neurodegenerative diseases, including Parkinson's disease, and provide additional support for a pathological activation of the ISR as a common theme in these pathologies.

In summary, our findings provide new insights into the requirement of Miro1 and Miro2 for mitophagy following mitochondrial damage and highlight their importance for mitochondrial homeostasis *in vitro* and *in vivo*. Moreover, we uncover a role of Miro1 as part of the Parkin receptor complex on the OMM and its potential role in tuning Parkin-mediated mitochondrial quality control. Long-term deletion of Miro1 in neurons *in vivo* leads to upregulation of Mfn1/2, remodelling of the mitochondrial network and induction of the ISR. This may open new strategies to target mitochondrial dysfunction in PD pathogenesis and other related diseases with alterations in the clearance of dysfunctional mitochondria.

# Materials and Methods

## Animals

The *Rhot1* (MBTN_ EPD0066_2_F01; Allele: *Rhot1*$^{tm1a(EUCOMM)Wtsi}$) and *Rhot2* (MCSF_ EPD0389_5_A05; *Rhot2*$^{tm1(KOMP)Wtsi}$) mice lines were obtained from the Wellcome Trust Sanger Institute as part of the International Knockout Mouse Consortium (IKMC) (Skarnes *et al*, 2011) and previously published (Lopez-Domenech *et al*, 2016). CAMKIIα-CRE strain has been described previously (Mantamadiotis *et al*, 2002). MitoDendra (B6;129S-Gt(ROSA) 26Sor$^{tm1(CAG-COX8a/Dendra2)Dcc}$/J) line was obtained from The Jackson Laboratory and was previously described (Pham *et al*, 2012).

PINK1-KO (EPD0787_2_G02; C57BL/6N-Pink1$^{tm1b(EUCOMM)Wtsi}$/H) mouse line was obtained from Harwell (UK). Animals were maintained under controlled conditions (temperature 20 ± 2°C; 12-h light–dark cycle). Food and water were provided *ad libitum*. All experimental procedures were carried out in accordance with institutional animal welfare guidelines and licensed by the UK Home Office in accordance with the Animals (Scientific Procedures) Act 1986. All data involving procedures carried out in animals are reported in compliance with ARRIVE guidelines (Kilkenny *et al*, 2010).

## DNA constructs

cDNA constructs encoding $^{myc}$Miro1, $^{myc}$Miro1$^{ΔEF}$ and MtDsRed have been previously described (Macaskill *et al*, 2009). $^{Myc}$Miro2 construct was previously published (Lopez-Domenech *et al*, 2016). $^{Flag}$Parkin cDNA was already published (Birsa *et al*, 2014) K153R, K182R, K187R, K194R, K572R and $^{myc}$Miro1$^{5R}$ (K153R, K182R, K187R, K194R and K572R) were made by site-directed mutagenesis on the $^{myc}$Miro1 backbone. The $^{myc}$Miro1$^{allR}$ construct, in which all lysine residues (except K612 and K616 to avoid mistargeted localisation) were replaced by arginine, was purchased from Life Technologies and was inserted into the pML5-myc backbone by restriction digestion. R572K $^{myc}$Miro1 was made by site-directed mutagenesis on the $^{myc}$Miro1$^{allR}$ backbone (and has all K residues mutated to R apart from K572). pRK5-HA-Ubiquitin was from Addgene [plasmid #17608, (Lim *et al*, 2005)]. $^{YFP}$Parkin was from Addgene [plasmid #23955, (Narendra *et al*, 2008)]. $^{GFP}$PINK1 construct was obtained from Addgene [plasmid #13316, (Beilina *et al*, 2005)].

## Antibodies

Primary antibodies for western blotting were as follows: mouse anti-Mfn1 (Abcam ab57602, 1:500), mouse anti-Mfn2 (NeuroMab 75-173, 1:500), mouse anti-Mfn2 (Abcam ab56889, 1:500), mouse anti-Parkin (Cell Signaling Technology 4211, 1:500), mouse anti-PINK1 mouse (NeuroMab, Clone N357/6, 1:3), mouse anti-VDAC1 (NeuroMab 75-204 1:1,000), mouse anti-ATP5a (from OXPHOS rodent antibody cocktail, Abcam ab110413, 1:2,000), rabbit anti-Actin (Sigma A2066 1:1000), rabbit anti-eIF2α (phospho S51) (Abcam ab32157, 1:500), rabbit anti-eIF2α (Cell Signaling Technology 9722, 1:500) and mouse anti-Ubiquitin (Enzo life sciences P4D1, 1:1,000). Primary antibodies for immunofluorescence were as follows: rat anti-GFP (Nacalai Tesque 04404-26, 1:2,000) and rabbit Anti-phospho-Ubiquitin (Ser65) (Merck Millipore ABS1513-I, 1:300), ApoTrack™ Cytochrome C Apoptosis WB Antibody Cocktail (CValpha, PDH E1alpha, cytochrome C, GAPDH) was from Abcam (1:1,000, mouse), anti-Actin (1:2,000, rabbit) and anti-Flag (1:1,000, rabbit) were from Sigma, and anti-Tom20 (1:500, rabbit) was from Santa Cruz Biotechnology. Anti-myc (9E10) and anti-HA were obtained from 9E10 and 12CA5 hybridoma lines, respectively, and used as supernatant at 1:100.

## Cell lines

$^{Flag}$Parkin stably overexpressing SH-SY5Y cells are a gift from Dr. Helen Ardley (Leeds Institute of Molecular Medicine) and were previously described (Ardley *et al*, 2003). WT and Miro$^{DKO}$ mouse embryonic fibroblasts (MEFs) were characterised previously (Lopez-

Domenech et al, 2018). Myo19KO RpE1 cells were generated using CRISPR/Cas9 technology. Briefly, two guide RNAs (5′caccGCCG AGTTAACCAGGAACGAA3′ and 5′ caccGCCACAGTCATCAAGCGT GCA 3′) were designed and cloned individually into px335BB cas9 (Addgene). Cell colonies originating from single clones were then sequenced and immunoblotted to confirm Myo19 depletion. Cells were maintained in Dulbecco's modified Eagle medium (DMEM) with 4,500 mg/l glucose and supplemented with foetal bovine serum (FBS) (SH-SY5Y and RpE1 10%; MEFs 15%), glutamax 2 mM, penicillin 120 µg/ml and streptomycin 200 µg/ml, and kept at 37°C and 5% $CO_2$ atmosphere.

### Biochemical assays

For ubiquitination assays, cells were lysed in RIPA buffer (50 mM Tris pH 7.4, 150 mM NaCl, 1 mM EDTA, 2 mM EGTA, 1% NP-40, 0.5% deoxycholic acid, 0.1% SDS, 1 mM PMSF and antipain, leupeptin, pepstatin at 10 µg/ml each). Lysates were then incubated on a rotating wheel at 4°C for 1 h, and then, nuclei and cellular debris were spun down at 20,000 g for 10 min. Supernatants were incubated at 4°C for 2 h with 10 µl of a 50% slurry of anti-myc beads (Sigma).

### Primary neuronal cultures

Primary cultures were prepared from E16.5 mice as in Lopez-Domenech et al (2016). Briefly, hippocampi from each embryo were dissected independently in ice-cold HBSS. Dissected tissue was treated with 0.075% trypsin in 1 mL HBSS per embryo at 37°C for 15 min. Tissue was washed twice and triturated to a single cell suspension. 350,000 cells were added to 6-cm dishes containing 8–10 PLL-coated glass coverslips in 5 ml attachment media (minimal essential medium, 10% horse serum, 1 mM sodium pyruvate and 0.6% glucose (Invitrogen)). Media was changed to maintenance media (Neurobasal plus B27 supplement (Invitrogen)) 6–12 h after plating. A tail section from each embryo was retained for genotyping.

### Lipofectamine transfection

Neurons were transfected at 7 DIV using Lipofectamine 2000 (Invitrogen). Per 6-cm dish, 1.5 µg DNA was combined with 300 µl Neurobasal (NB) and 2 µl Lipofectamine with 300 µl NB. After 5 min, Lipofectamine was added to the DNA and incubated for 30 min to complex. 1.4 ml of warm NB containing 0.6% glucose was added to the complex. 2 ml total volume was added directly to 6-cm dish and incubated for 90 min at 37°C. Solution was then aspirated and replaced with conditioned media. Cultures were left to rest for > 72 h prior to experimentation.

### Tissue processing

Animals of the selected age were culled by cervical dislocation or $CO_2$ exposure. Brains were snap-frozen in liquid nitrogen and lysates produced with the appropriate lysis buffers. For histological studies, animals were deeply anesthetised and transcardially perfused with chilled 4% PFA to maintain mitochondrial integrity. Brains were further fixed by immersion in 4% PFA overnight at 4°C, cryoprotected in 30% sucrose-PBS for 24–48 h and stored at −80°C.

Tissue was serially cryosectioned in a Bright OTF-AS Cryostat (Bright Instrument, Co. Ltd.) at 30 µm thickness and stored in cryoprotective solution (30% PEG, 30% glycerol in PBS) at −20°C until used.

### Immunohistochemistry

Immunohistochemistry studies were performed in free-floating sections. Briefly, tissue sections were washed 3–5 times in PBS over 30 min and permeabilised in PBS with 0.5% Triton X-100 over a total of 3–5 washes during 30 min. Sections were then blocked at room temperature (RT) in a solution containing 3% BSA, 10% foetal bovine serum and 0.2 M glycine in PBS with 0.5% Triton X-100 for 3–4 h. Eventually, tissue was further blocked overnight at 4°C in the same blocking solution plus purified goat anti-mouse Fab-fragment (Jackson ImmunoResearch) at a concentration of 50 µg/ml to reduce endogenous background. Sections were then further washed and incubated overnight at 4°C in primary antibodies prepared in blocking solution. After washing in PBS with 0.5% Triton X-100, secondary antibodies were applied in blocking solution and incubated at RT for 3–4 h. Sections were mounted on glass slides using Mowiol mounting media and stored at 4°C in the dark until documented.

### Western blotting

For western blotting, 20 µg of protein from lysates of adult hippocampus or from cultured MEF cell lines was loaded on 8–12% acrylamide gels and transferred to nitrocellulose membranes (GE Healthcare Bio-Sciences) using the Bio-Rad system. Membranes were blocked in 4% powder skimmed milk in TBS-T for 1–2 h. Antibodies were incubated in blocking solution (overnight at 4°C for primary antibodies or 1 h at RT for HRP-conjugated secondary antibodies). Membranes were developed using the ECL-Plus reagent (GE Healthcare Bio-Sciences) and acquired in a chemiluminescence imager coupled to a CCD camera (ImageQuant LAS 4000 mini). Densitometric analysis was performed using ImageJ software (https://imagej.nih.gov/ij/).

### Immunocytochemistry

Cells growing in coated coverslips or hippocampal cultures were fixed when required in 4% PFA for 10 min at RT and rinsed several times in PBS. For immunocytochemistry, coverslips were permeabilised in PBS with 0.1% Triton X-100 and incubated for 1 h in blocking solution (1% BSA, 10% foetal bovine serum, 0.2 M glycine in PBS with 0.1% Triton X-100). Primary antibodies where applied in blocking solution at the desired concentration and incubated for 2 h. Alexa Fluor (Invitrogen) secondary antibodies were incubated for 1 h at 1:800 in blocking solution. Coverslips were mounted in Mowiol mounting media and kept in the dark at 4°C until imaged.

### Mitophagy assays

[YFP]Parkin- and MtDsRed-expressing neurons were used for experimentation at 10–12 DIV. Cultures were treated with 1 µM valinomycin (Sigma) diluted in neuronal maintenance media for various time points ranging from 30 min to 5 h prior to fixation with 4% PFA solution.

WT and Miro^DKO MEFs (and WT and Myo19^KO RpE1 cells) were transfected with ^YFPParkin or co-transfected with ^YFPParkin (or ^GFPPINK1 in PINK1 recruitment experiments) and ^mycMiro1, ^mycMiro2, ^mycMiro1^5R or ^mycMiro1^allR when required. The next day, cells were split and seeded onto fibronectin-coated coverslips at 20 µg/ml at a density of 50,000 cells/cm². 24 h later, mitophagy was induced by 10 µM FCCP treatments that were applied for the selected time points (1, 3, 6, 12 and 24 h in Parkin recruitment experiments, or 15, 30 or 60 min in PINK1 recruitment experiments). Coverslips were then fixed and stored at −20°C until processed for immunofluorescence.

### Analysis of mitochondrial distribution using micropatterns

WT and Miro^DKO MEF cells were transfected with ^mycMiro1, ^mycMiro2 or the ubiquitination mutants ^mycMiro1^5R or ^mycMiro1^allR. 24 h after transfection, cells were seeded onto "Y"-shaped micropatterned coverslips coated with fibronectin (CYTOO) at 15,000–20,000 cells/cm². Cells were allowed to attach to the permissive substrate for 4 h and then incubated with MitoTracker-Orange for 40 min and fixed with PFA 4%. Fixed cells were immunostained to identify the transfected cells with an anti-myc antibody, and selected cells were imaged in a confocal microscope using sub-saturation parameters for the MitoTracker channel. Sholl analysis of mitochondrial signal distribution was performed using a custom-made ImageJ plugin (Lopez-Domenech *et al*, 2018). Mitochondrial signal was quantified within shells radiating out from the soma at 1-µm intervals. The cumulative distribution of mitochondrial signal as a function of the distance from the centre of the cell to the periphery was normalised per each cell. An average value was calculated for each distance interval for all the cells in the experiment, and a final profile or mitochondrial probability map (MPM) was plotted per condition. In all plots, a theoretical profile (grey dotted line) of a homogeneously distributed signal was generated from a GFP-overexpressing cell. The distance at which 95% of the total mitochondrial mass is found (Mito^95 value) was calculated per each cell by interpolation. One average Mito^95 value was calculated per condition using total number of cells as statistical "$n$" ($n$ = number of cells). The whole experiment was repeated at least three times.

### Quantitative PCR

To quantify mitofusin1 and mitofusin2 gene expression, we performed quantitative PCR (qPCR). One hippocampus per animal was carefully dissected, and RNA was extracted and treated with DNase I (Amplification grade; Thermo Fisher Scientific) to remove any remaining trace amounts of DNA. cDNA was generated with 20 ng of RNA by using the Qiagen Whole Transcriptome Amplification Kit as described in the manufacturer's protocol. Primers for qPCR were designed by PrimerBank (Massachusetts General Hospital, Boston, US) and are as follows: Gapdh (GenBank accession NM_008084; forward ATGACATCAAGAAGGTGGTG; reverse CATACCAGGAAATGAGCTTG), Hprt (GenBank accession NM_013556; forward GTTGGATACAGGCCAGACTTTGTTG; reverse GAGGGTAGGCTGGCCTATAGGCT), Mfn1 (GenBank accession NM_024200; forward CCTACTGCTCCTTCTAACCCA; reverse AGGGACGCCAATCCTGTGA) and Mfn2 (GenBank accession NM_133201; forward CCAACTCCAAGTGTCCGCTC; reverse GTCCAGCTCCGTGGTAACATC).

qPCRs were performed by using Sybr Green reagent (Merck, UK) on a CFX96 Real-Time System (Bio-Rad, Hercules, CA). PCR conditions were 94°C for 2 min, followed by 40 three-step cycles of 94°C for 15 s, 60°C for 30 s and 72°C for 30 s. Gapdh and Hprt were used as housekeeping gene controls. Primers were validated by PCR and agarose gel electrophoresis and had similar amplification efficiencies when validated by using a serial dilution of representative cDNA. Samples were obtained from 4 different WT and 4 different Miro1^CKO animals which were considered the statistical "$n$". The experiment was repeated three times, and in each experiment, samples were amplified in triplicate. Relative quantification was determined according to the delta-delta c(t) method (Livak & Schmittgen, 2001).

### Image acquisition and analysis

#### Confocal imaging

Confocal images (1,024 × 1,024 pixels) were acquired on a Zeiss LSM700 upright confocal microscope (Carl Zeiss, Welwyn Garden City, UK) using a 40× or 63× oil immersion objective (NA: 1.3 and NA: 1.4, respectively). Images were processed with ImageJ software (http://imagej.nih.gov/ij/). Stages of Parkin-induced mitophagy were scored from images taken with the 40× objective by a blinded researcher. Data were collected from at least three independent experiments. For histology studies, confocal images were taken using a 5× (NA: 0.16) or 10× (NA: 0.3) air objective or a 63× oil immersion objective (NA: 1.4). Acquisition parameters were kept constant over experimental conditions. All histology experiments were performed with age-matched animals. For high-resolution imaging of primary hippocampal neurons and mitoDendra-expressing brains, a 63× oil immersion objective (NA: 1.4) coupled to a Zeiss LSM 880 inverted confocal microscope with Airyscan technology was used.

#### Electron microscopy

Brain slices were immersed and fixed with 2% paraformaldehyde, 2% glutaraldehyde and 2% sucrose in 0.1 M cacodylate buffer pH 7.3 and then post-fixed in 1% OsO$_4$/0.1 M cacodylate buffer pH 7.3 at 3°C for one and a half hours. Sections were then *en bloc* stained with 0.5% uranyl acetate dH$_2$0 at 3°C for 30 min. Specimens were dehydrated in a graded ethanol-water series, infiltrated with Agar-100 resin and then hardened at 60°C for 24 h. 1-µm sections were cut and stained with 1% toluidine blue in dH$_2$0 for light microscopy. At the correct position, ultra-thin sections were cut at 70–80 nm using a diamond knife on a Reichert ultra-microtome. Sections were collected on 300 mesh copper grids and then stained with lead citrate. Sections were viewed in a Joel 1010 transition electron microscope and images recorded using a Gatan Orius camera.

## Data availability

All data supporting the findings of the present study are available from the corresponding author upon reasonable request.

**Expanded View** for this article is available online.

## Acknowledgements

This work was supported by an ERC starting grant (282430) and Lister Institute for Preventive Medicine prize to J.T.K. G.L-D. is supported by AstraZeneca

Pharmaceuticals post-doctoral programme (000033622/1363780). J.H.H. is a recipient of a BBSRC (BB/P504865/1) Case Award. C.C-C receives funding from the Medical Research Council (1368635). J.V.P. was supported by Cancer Research UK (C1529/A17343).

## Author contributions

GL-D, JHH and JTK conceived the project. GL-D, JHH, NB and JTK designed experiments. GL-D, JHH, CC-C, NB and CM performed experiments and analysed the data. JVP generated the Myo19[KO] RpE1 cells. RB, DC and NJB provided critical input and support. GL-D, JHH and JTK wrote the paper. RB, DC, NJB and JTK supervised the project.

## Conflict of interest

G.L-D. is supported by AstraZeneca Pharmaceuticals post-doctoral programme (000033622/1363780). R.B., D.C. and N.J.B. are or have been employees of AstraZeneca Pharmaceuticals.

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
