## [Review Process File · The EMBO Journal]

Loss of neuronal Miro1 disrupts mitophagy and induces hyper-activation of the integrated stress response

Guillermo López-Doménech, Jack Howden, Christian Covill-Cooke, Corinne Morfill, Jigna Patel, Roland Burli, Damian Crowther, Nicol Birsa, Nicholas Brandon, and Josef T Kittler
DOI: [10.15252/embj.2018100715](https://doi.org/10.15252/embj.2018100715)

Corresponding author(s): Josef Kittler (j.kittler@ucl.ac.uk)

Review Timeline:

Submission Date:	14th Sep 18
Editorial Decision:	13th Jan 19
Revision Received:	19th Jan 21
Editorial Decision:	22nd Feb 21
Revision Received:	23rd Apr 21
Accepted:	3rd May 21

Editor: Elisabetta Argenzio

Transaction Report:

Thank you for submitting your manuscript entitled "Miro ubiquitination is critical for efficient damage-induced PINK1/Parkin-mediated mitophagy" (EMBOJ-2018-100715) to The EMBO Journal. I have now read your point-by-point response and discussed it with other members of the editorial team.

The outcome is that we find the proposed plan to address the referees' concerns reasonable. Given the overall interest of your study, I would like to invite you to revise the manuscript as requested in the referee's reports. In addition, I ask you to discuss your findings in the context of the most recent literature. I would like to point it out that addressing all the referees' points would be essential to consider publication in The EMBO Journal and that we need strong support from the referees for publication here.

Please note that it is The EMBO Journal policy to allow only a single major round of revision and that it is therefore important to resolve the main concerns at this stage.

When preparing your letter of response to the referees' comments, please bear in mind that this will form part of the Review Process File, and will therefore be available online to the community. For more details on our Transparent Editorial Process, please visit our website:

http://emboj.embopress.org/about#Transparent_Process

We generally recommend three months as standard revision time. As a matter of policy, competing manuscripts published during this period will not negatively impact on our assessment of the conceptual advance presented by your study. However, please contact me as soon as possible upon publication of any related work in order to discuss how to proceed.

Thank you again for the opportunity to consider this work for publication, and please feel free to contact me with any questions about submission of the revised manuscript to The EMBO Journal. I look forward to your revision.

Referee #1:

Miro proteins facilitate mitochondrial transport by regulating the activity of kinesin/dynein motors in coordination with myosin-19. The current study by López-Doménech et al examined the potential role of Miro proteins for PINK1/Parkin-dependent mitochondrial degradation.

The study shows that loss of both Miro1 and 2 proteins in mouse embryonic fibroblasts (MEFs) of double knockouts (MiroDKO) impairs translocation of the ubiquitin ligase Parkin to mitochondria and slows mitophagy indicating that Miro is required for efficient progression of mitophagy. Deletion of Miro1 in cultured neurons had a similar effect. Ubiquitination assays revealed that after mitochondrial damage Miro is ubiquitinated on multiple lysine residues by Parkin. Mutating 5 or all of the potential ubiquitination sites of Miro suppressed Miro degradation after inducing mitochondrial damage and slowed mitophagy. Rescue experiments re-expressing either WT Miro1 indicate that ubiquitination of Miro1 is a required step for efficient mitophagy. Finally, conditional knockout of Miro1 impaired mitochondrial homeostasis and caused an age-dependent upregulation of Mfn1 and

Mfn2.

The study is well executed and significantly advances our understanding of mechanisms underlying PINK1/Parkin-mediated mitophagy. I have, however, some concerns.

- 1) I am wondering whether the effect of the Miro ubiquitination mutations "5R" and "allR" on mitophagy is an indirect consequence of a failure of mitochondrial transport. Hence, the authors should show whether these mutations affect mitochondrial transport in any significant manner.
- 2) The authors claim that mutations of any one of the potential ubiquitination sites does not significantly alter Miro1 protein levels (Fig 2C). Please provide a quantification since "allR" and R527" mutants appear to be reduced on the shown blot.
- 3) page 8-end. "Unexpectedly, however, we also noted that the damage-induced loss of PDHE1 α (a mitochondrial matrix protein) was reduced in both Miro1 5R and Miro1 allR expressing cells (Fig 2I), while Parkin levels also appeared to be stabilized (Fig 2F and S1)". I am confused, fig 2I shows that PDHE1 α is stabilized at 3h of FCCP treatments in 5R and allR but not after 6h. Furthermore, Parkin levels appear to be stabilized only for the allR mutation after 3h. Please clarify.
- 4) Please indicate the timepoint after FCCP application for the shown images in Fig 1A and the respective genotype (I assume its WT). Please indicate hours (h) on the x-scale of graphs shown in Fig. 1B-E.
- 5) Fig 2H-I: the labeling of the y-axis is confusing; it should be % myc-Miro levels and % PDH E1 α levels.

Referee #2:

The study by Dr. Kittler and colleagues investigates the role of Miro and its ubiquitination in PINK1 and Parkin mediated mitophagy in cells in culture and in vivo in mice. Consistent with the literature they find Miro is promiscuously ubiquitinated at multiple lysine residues. Moreover, they find combined knockdown of Miro1/2 or block of their ubiquitination leads to delayed Parkin translocation onto damaged mitochondria and reduced clearance. Further they suggest that postnatal knockout of Miro1 in vivo in the hippocampus and cortex disrupts mitophagy and leads to an age dependent upregulation of mitofusins and enlarged, hyperfused mitochondria.

While the increase in mitofusin levels and abnormal mitochondrial morphology and size in Miro conditional knockouts are certainly interesting, the findings remain largely unexplained. Overall, the study appears rather descriptive and as such somewhat premature. Given that the mechanisms remain unclear, the study does not provide significant advances as is.

Other specific comments:

The number of cells scored and number of experiment repeats are not always clear, but should be stated. The statistical test should be two-way ANOVA for all comparisons across genotypes and over time.

To exclude any potentially confounding effect, it is key to confirm similar baseline and FCCP

responses (depolarization and accumulation of PINK1 protein) in DKO vs WT cells as well as for rescue experiments. Also it should be assessed if differences in Parkin expression levels affect the kinetics (e.g. Fig 1 ii clearly shows less YFP-parkin or are these images from different exposure times).

Besides their mitochondrial localization, are the Miro KR mutant versions functional with respect to e.g. mitochondrial transport?

Figure 3: How was mitochondrial turnover assessed?

Miro KR seems to similarly affect Parkin translocation than loss of Miro, which will translate into reduced turnover overall. Unclear how the two suggested Miro functions can be distinguished in this set-up. It would be easier to understand if the early stages were not impaired in Miro KR mutants and only later stages would be impaired (delay of Parkin translocation vs reduced mitophagy).

Figure 4: H) legend appears to be missing.

Figure 5: Is the upregulation of mitofusins mediated on the transcriptional or only on protein level? Are hyperfused mitochondria a result of enhanced fusion or also downregulation of the fission machinery? Are Opa1 levels affected too?

Is there some compensation between both Miro proteins when one is knocked down/out? It would be important to distinguish between compensatory effects, enhanced mitophagy activation, or reduced turnover.

Figure S1: It is unclear what has been measured here. Parkin is (auto)ubiquitinated and moves into higher molecular weight fractions at some point not detectable. Is the same seen in Miro double KO cells?

Response to Editors comments

Thank you for submitting your manuscript on a role for Miro proteins in Parkin-mediated mitophagy to The EMBO Journal. Your study has been sent to two referees for evaluation, and we have now received reports from them, which are enclosed below.

As you will see, while both referees consider the findings to be potentially interesting, they also raise major criticism on the study. In particular, they request you to test the mitochondrial transport of Miro ubiquitination mutants as well as to quantify their protein levels (point 2 from referee #1). Referee #2 is the most critical one. S/he states that the study lacks mechanistic insight and that the main findings remain largely unexplained at this stage. In addition, this referee points out that additional data are required to explain the upregulation of mitofusins in your model and that compensation between Miro proteins have to be investigated in depth.

These are important points in our view, and given the substantial criticisms raised regarding conclusiveness and lack of mechanistic insight, we find it difficult to commit to going further with this manuscript in The EMBO Journal. However, I would offer you the chance to read the reports and to let us know about your view on the critique and how the concerns raised by the referees could be addressed within the time-frame of a revision. We would then take that into consideration - and possibly also reconsult with the referees to determine if such a revision would address their concerns - before making a final decision on your study. I would like to stress that we need strong endorsement from the referees in order to fully commit to a revised manuscript.

Please feel free to contact me with any questions related to this matter. By conducting this exchange at the current stage, I hope to avoid inviting a revision with a high risk of being rejected by the referees following extensive experimental efforts on your side.

I look forward to hearing from you.

We are very grateful for your patience and for giving us the opportunity to respond to the referees' helpful comments on our manuscript (EMBOJ-2018-100715), the majority of which we have addressed (see also the point-by-point response provided below) and which we agree have further improved the paper. We note that both reviewers find several aspects of the work interesting and of importance to the field.

We find Referee#1 to be broadly supportive of our work and while they raise some important points, we do not feel that they raised any major criticisms. As you point out, Referee#1 notes that we should further determine the expression levels of Miro ubiquitination mutants and their impact on mitochondrial trafficking (a point also raised by Referee#2). We completely agree and have provided evidence that the effect of Miro1 in the regulation of mitophagy is independent on its ability to regulate mitochondrial trafficking. We show that re-expression of Miro1^{5R}, a construct that shows delayed Parkin recruitment and mitochondrial clearance, successfully rescues mitochondrial distribution in Miro^{DKO} cells to levels similar to control Miro1^{WT}. This demonstrates that defects in

ubiquitination in Miro1 are linked to the dysregulation of mitophagy independently of its role in mitochondrial trafficking and distribution.

Referee#2 additionally raises concerns that our study currently 'lacks mechanistic insight'. We would respectfully disagree with this criticism of Referee#2 which we feel is rather unfair given that our study already provides several novel insights into the role of Miro proteins in mitophagy:

i) We identify a key role for Miro proteins and their ubiquitination in the recruitment of Parkin during the initiation of the mitophagic process.

ii) We provide important information regarding the critical need of Miro1 degradation after mitochondrial damage for the progression of mitophagy.

iii) Importantly, we also show that the long-term consequences of altered mitophagy upon Miro1 disruption is the dysregulation of mitochondrial dynamics and mitochondrial homeostasis *in vivo*, in mature neurons from intact brains, providing a mechanistic link between the dysregulation of mitochondrial dynamics and neurodegeneration *in vivo*. Although recent studies have linked Miro1 dysregulation to Parkinson's disease pathology using *in vitro* models (Berenguer-Escuder, Grossmann et al., 2020, Grossmann, Berenguer-Escuder et al., 2019, Hsieh, Li et al., 2019, Hsieh, Shaltouki et al., 2016), a major limitation of this cell culture work is that it can only assess the short-term (days to weeks) impact of Miro1 dysregulation for mitophagy and cellular physiology. In contrast, our work highlights the long term (up to 1 year) consequences of such dysregulations for aged neurons in intact brains, offering invaluable insight into the cellular consequences of long-lasting dysregulation of mitochondrial homeostasis associated with human disease.

Thus, our study already adds important mechanistic understanding of the mitophagic process and the consequences of its disruption at several levels. Notwithstanding, we acknowledge that both you and Referee#2 do raise a good point, which we interpret as being that the manuscript would be further strengthened if we could provide additional insight. Our mouse model provides an unmatched opportunity to explore in more detail the long-lasting consequences of disrupting mitophagy *in vivo* and the significance of increased mitofusins levels and disruption of mitochondrial dynamics for neuronal survival and death. Importantly, we now provide a further link between the increased mitofusin expression and dysregulation of mitochondrial dynamics we observe and the consequences for neuronal health and survival. We now show high levels of eIF2 α phosphorylation in our Miro1 conditional knockout model, which provides a link between the sustained activation of the Integrated Stress Response and the neurodegenerative phenotype of our model.

We also enclose a point-by-point response to the comments of both referees where we address their concerns.

We believe we have a strong endorsement from Reviewer 1, and we have addressed Referee#2 concerns. In addition, we provide a deeper insight into the long-term consequences of disrupting Miro1-dependent mitophagy in intact brains including a mechanistic link between the dysregulation of mitochondrial dynamics and neurodegeneration *in vivo*. We very much hope you will now find our revised manuscript suitable for publication in The EMBO Journal.

A point-by-point response to the referees' comments is provided below:

Referee #1

Miro proteins facilitate mitochondrial transport by regulating the activity of kinesin/dynein motors in coordination with myosin-19. The current study by López-Doménech et al examined the potential role of Miro proteins for PINK1/Parkin-dependent mitochondrial degradation.

The study shows that loss of both Miro1 and 2 proteins in mouse embryonic fibroblasts (MEFs) of double knockouts (MiroDKO) impairs translocation of the ubiquitin ligase Parkin to mitochondria and slows mitophagy indicating that Miro is required for efficient progression of mitophagy. Deletion of Miro1 in cultured neurons had a similar effect. Ubiquitination assays revealed that after mitochondrial damage Miro is ubiquitinated on multiple lysine residues by Parkin. Mutating 5 or all of the potential ubiquitination sites of Miro suppressed Miro degradation after inducing mitochondrial damage and slowed mitophagy. Rescue experiments re-expressing either WT Miro1 indicate that ubiquitination of Miro1 is a required step for efficient mitophagy. Finally, conditional knockout of Miro1 impaired mitochondrial homeostasis and caused an age-dependent upregulation of Mfn1 and Mfn2.

The study is well executed and significantly advances our understanding of mechanisms underlying PINK1/Parkin-mediated mitophagy. I have, however, some concerns.

We thank the reviewer for stating that our study is well executed and significantly advances our understanding of mechanisms underlying PINK1/Parkin-mediated mitophagy

1) I am wondering whether the effect of the Miro ubiquitination mutations "5R" and "allR" on mitophagy is an indirect consequence of a failure of mitochondrial transport. Hence, the authors should show whether these mutations affect mitochondrial transport in any significant manner.

We agree with the referee that this is an important point. We have used an adhesive micropatterned substrate based analysis of mitochondrial distribution in MEF cell lines for which we have extensive experience (Covill-Cooke, Toncheva et al., 2020, Lopez-Domenech, Covill-Cooke et al., 2018, Modi, Lopez-Domenech et al., 2019). While we find that re-expression of our non-ubiquitinable Miro1^{allR} construct cannot rescue mitochondrial distribution in Miro^{DKO} cells we show that re-expression of Miro1^{5R}, rescues mitochondrial distribution in Miro^{DKO} to a level similar to that of Miro1^{WT}. Importantly, Miro1^{5R} shows a delayed recruitment of Parkin to mitochondria and, most importantly, delayed mitochondrial clearance under mitochondrial damage. From these experiments we conclude that the effect on mitophagy of these constructs is not directly dependent on their ability to regulate mitochondrial transport but rather on the ability to interact with and recruit Parkin to damaged mitochondria. We discuss this new data accordingly throughout the result and discussion sections.

2) The authors claim that mutations of any one of the potential ubiquitination sites does not significantly alter Miro1 protein levels (Fig 2C). Please provide a quantification since "allR" and R527" mutants appear to be reduced on the shown blot.

As requested by the referee we provide here (see reviewers' Fig 1 below) a quantification of the expression levels of all constructs. We find that only two of our constructs (K572R and K182R) exhibit a significant decrease in expression (of approximately 25-35%). For one of these, the same K572R, was used in a previously published paper (Safiulina, Kuum et al., 2019). Importantly, the levels of the main constructs in which we base our study (principally the ^{myc}Miro1^{5R} construct) does not appear to be

significantly changed, supporting that the levels of expression of our constructs are not behind the effects of Miro1 in mitophagy that we describe throughout our manuscript.

Reviewers' Fig 1: Quantification of protein levels of all Miro1 ubiquitination mutants shown in Figure 2C. While there is a significant 25-35% reduction in the expression of 2 of them, the rest of the mutants express at levels similar the Miro1^{WT}, control construct. Importantly, the Miro1^{5R} construct does not show significant differences compared to Miro1^{WT} expression (data collected from 3 different expression experiments (n=3); One way ANOVA with Newman-Keuls poshoc analysis)

3) page 8-end. "Unexpectedly, however, we also noted that the damage-induced loss of PDHE1 α (a mitochondrial matrix protein) was reduced in both Miro1 5R and Miro1 allR expressing cells (Fig 2I), while Parkin levels also appeared to be stabilized (Fig 2F and S1)".

I am confused, fig 2I shows that PDHE1 α is stabilized at 3h of FCCP treatments in 5R and allR but not after 6h. Furthermore, Parkin levels appear to be stabilized only for the allR mutation after 3h. Please clarify.

We have now reworded this section to improve the clarity of the message. We show that defects in Miro1 ubiquitination are associated with an important delay in the mitophagic process although the mitophagic process is not completely stopped. We have included a more accurate description of the stabilization of PDHE1 α and Parkin levels by the Miro1 ubiquitination mutants.

4) Please indicate the timepoint after FCCP application for the shown images in Fig 1A and the respective genotype (I assume its WT). Please indicate hours (h) on the x-scale of graphs shown in Fig. 1B-E.

We have now included in the figure the time-points after FCCP treatment. We have also included in the figure legend the genotype of the cells shown in the images. As this referee correctly pointed out they are all examples of WT cells. We have also included a title in the x-axis of all graphs presented in Figure 1.

5) Fig 2H-I: the labeling of the y-axis is confusing; it should be % myc-Miro levels and % PDH E1 α levels.

We are sorry for the wrong labelling. We have now corrected them as suggested

Referee #2

The study by Dr. Kittler and colleagues investigates the role of Miro and its ubiquitination in PINK1 and Parkin mediated mitophagy in cells in culture and in vivo in mice. Consistent with the literature they find Miro is promiscuously ubiquitinated at multiple lysine residues. Moreover, they find combined knockdown of Miro1/2 or block of their ubiquitination leads to delayed Parkin translocation onto damaged mitochondria and reduced clearance. Further they suggest that postnatal knockout of Miro1 in vivo in the hippocampus and cortex disrupts mitophagy and leads to an age dependent upregulation of mitofusins and enlarged, hyperfused mitochondria.

While the increase in mitofusin levels and abnormal mitochondrial morphology and size in Miro conditional knockouts are certainly interesting, the findings remain largely unexplained. Overall, the study appears rather descriptive and as such somewhat premature. Given that the mechanisms remain unclear, the study does not provide significant advances as is.

We would respectfully disagree with the general criticism that our study does not provide significant mechanistic insight. The identification of the importance of Miro for facilitating the recruitment of Parkin to mitochondria is by itself an important novel insight. Since our submission a recent report independently confirmed our result (Safiulina et al., 2019) which supports the importance of our observation. Furthermore, our work provides additional mechanistic insight by showing that the ubiquitination of Miro1 plays a critical role in the facilitation of Parkin recruitment. Finally, we further show that the role of Miro1 in the regulation of mitophagy is not limited to the recruitment of Parkin to damaged mitochondria but also, that Miro1 must be degraded after its ubiquitination in order to allow the progression of mitophagy and final mitochondrial clearance.

Moreover, given that the vast majority of studies on the mitophagy pathway in mammals are performed in highly reductionist cell line preparations, we think the additional insights we provide *in vivo*, in aged neurons from intact brains are important in their own right. In the revised version of our manuscript, we have now built on this further and have complemented our study with a deeper characterization of our mouse model. We have explored in more detail the long-lasting consequences of disrupting mitophagy *in vivo* and the significance of increased mitofusin levels and disruption of mitochondrial dynamics for neuronal survival and death. To this end, we provide solid evidence that in a neuronal system the disruption of mitophagy associates with long term changes in mitofusin expression levels and drastic changes in mitochondrial dynamics. This severe dysregulation of mitochondrial dynamics leads to the hyperactivation of the integrated stress response which has been widely implicated in the neurodegenerative outcome of many pathological processes and that may explain the neurodegeneration phenotype associated to human disease caused by the dysregulation of mitophagy.

We hope that the additional insight provided in the revised version of the manuscript helps to address this referee concerns about the depth of our study.

Other specific comments:

The number of cells scored and number of experiment repeats are not always clear, but should be stated. The statistical test should be two-way ANOVA for all comparisons across genotypes and over time.

We now provide additional information regarding number of experiments, number of cells, replicates and the “n” used for statistical analysis for each experiment that can be found in the corresponding figure legend.

To exclude any potentially confounding effect, it is key to confirm similar baseline and FCCP responses (depolarization and accumulation of PINK1 protein) in DKO vs WT cells as well as for rescue experiments.

We have conducted additional experiments to quantify PINK1 recruitment and stabilization in the mitochondria in our different cell lines in basal conditions and under mitochondrial damage. We show that Miro deletion does not affect the timely stabilization of PINK1 on the mitochondria under damage. This supports the idea that the effect of Miro1 on mitophagy is through the recruitment of Parkin to damaged mitochondria. This new data can now be seen in Figure 1. In addition, we have also studied how the different expression mutants may affect the mitochondrial membrane potential and show that none of the constructs used has a significant effect (see Supplementary Figure 3).

Also it should be assessed if differences in Parkin expression levels affect the kinetics (e.g. Fig 1 ii clearly shows less YFP-parkin or are these images from different exposure times).

In our experiments, the extent to which Parkin is translocated to mitochondria in the different genotypes is not dependent on Parkin levels. We provide below (Reviewers' Fig 2) a correlation analysis showing that higher Parkin-expressing cells are not more prone to translocate Parkin to the mitochondria under mitochondrial damage.

Reviewers' Fig 2: Correlation analysis between total Parkin expression levels measured by average fluorescence intensity of transfected cells and the quantity of Parkin signal (Integrated Density) translocated to the mitochondria in WT cells overexpressing YFP-Parkin and treated with FCCP for 6 hours. Two different confocal images were taken from each cell. The first image was taken using the same acquisition parameters for all cells (in the subsaturation range) and was used to calculate and compare total Parkin levels. The second image was taken with specific parameters for each cell to achieve the same level of saturation signal (approximately 1-2% of cellular pixels were saturated) and was used to calculate the proportion of the total fluorescence translocated to mitochondria (Integrated density in ImageJ). A total of 23 cells were imaged from 2 independent experiments.

Importantly, we now also provide further evidence that the loss of Miro1 disrupts the kinetics of PINK1/Parkin mitophagy in the absence of exogenously expressed Parkin. We show in primary neurons, expressing endogenous levels of Parkin, that deletion of Miro1 impacts the amount of phospho-ubiquitin accumulation after mitochondrial damage with the use of an antibody specific to s65-phospho-ubiquitin (Figure 4G and H).

Besides their mitochondrial localization, are the Miro KR mutant versions functional with respect to e.g. mitochondrial transport?

As per referee #1 (point #1), we have performed mitochondrial distribution assays using an adhesive micropatterned substrate based analysis experiment (Covill-Cooke et al., 2020, Lopez-Domenech et al., 2018, Modi et al., 2019) to test whether Miro ubiquitin mutations are able to rescue mitochondrial distribution defects in Miro^{DKO} MEFs. We show that while the non-ubiquitinable Miro1^{allR} construct cannot rescue mitochondrial distribution in Miro^{DKO} cells, re-expression of Miro1^{5R}, which also shows a delayed Parkin translocation and mitochondrial clearance under mitochondrial damage, rescues mitochondrial distribution to the same extent as re-expression of the control, Miro1^{WT} construct. We conclude that the effect on mitophagy of these constructs is not directly dependent on their ability to regulate mitochondrial distribution.

Figure 3: How was mitochondrial turnover assessed?

To assess mitochondrial turnover, we quantified the number of cells that after 24 hours of FCCP treatment had lost all their mitochondrial mass as observed by loss of Tom20 signal. We have now clarified this in the manuscript

Miro KR seems to similarly affect Parkin translocation than loss of Miro, which will translate into reduced turnover overall. Unclear how the two suggested Miro functions can be distinguished in this set-up. It would be easier to understand if the early stages were not impaired in Miro KR mutants and only later stages would be impaired (delay of Parkin translocation vs reduced mitophagy).

We agree with the referee that this point would benefit from clearer presentation. Our experiments in Miro^{DKO} MEFs and in Miro1^{KO} neurons in Figures 1 and 2 show that the timely recruitment of Parkin to damaged mitochondria requires the presence of Miro1 in the mitochondria. This is further supported by our rescue experiments in Miro1^{KO} neurons under basal conditions, in which the over-expression of Miro1^{WT} and Miro1^{5R} recruited Parkin to mitochondria even in the absence of mitochondrial damage. In addition to this, we have shown that the ubiquitination of Miro1 is also a critical step required to amplify Parkin recruitment and stabilization under damage induced conditions. The fact that the poorly ubiquitinated Miro1^{5R} can still recruit Parkin under basal conditions (Fig 4I) but still shows a delay in Parkin recruitment under mitochondrial damage (Fig 3A, B and C and Fig 4J) highlights the requirement of Miro1 ubiquitination for Parkin recruitment during early stages of mitophagy. Indeed, using PINK1^{KO} neurons where Parkin dependent mitophagy is blocked, we further show that the Miro1-dependent recruitment of Parkin to healthy mitochondria is distinct from damage-induced Parkin recruitment/amplification.

Furthermore, with our experiments we characterize an additional effect caused by the mutation of Miro1 lysine residues. It is well known that Miro1 ubiquitination by Parkin leads to a very rapid degradation of Miro1 by the proteasome. Interestingly, we report here that blocking Miro1 ubiquitination results in the stabilization of Miro1 in the mitochondria (Fig 2F and G and Fig 3D and E). Miro1 stabilization also led to a delayed progression through mitophagy as seen by the stabilization of mitochondrial markers (Fig 2F and H) and a reduction in mitochondrial clearance (Fig 3E and F). Importantly, the delay in mitophagy progression observed with the expression of the Miro1 lysine mutants is significantly greater than that caused by the loss of Miro (Fig 3E and F), indicating that Miro1 stabilization interferes with mitophagy progression in addition to Miro1 being itself an important receptor for Parkin translocation. It has been suggested that the rapid ubiquitination and degradation of mitofusin 2 in response to mitochondrial damage could act as a gating mechanism, allowing the disassembly of the ER and mitochondria interactions and thus facilitating widespread Parkin ubiquitination of other mitochondrial substrates and rapid mitophagy progression (McLelland et al., 2018). In addition, expression of PTP51, an ER/Mitochondrial anchoring protein, results in the

suppression of Parkin-mediated mitophagy, supporting a key role for the stability of the ER/mitochondrial contacts sites (ERMCS) in the regulation of mitophagy progression (Puri, Cheng et al., 2019). Similar to mitofusin 2, Miro1, which is also rapidly ubiquitinated and degraded after mitochondrial damage, has also been shown to be an important regulator of ER-Mitochondria contact sites (ERMCS) (Modi et al, 2019). Therefore, we propose that Miro1 is a good candidate to be involved in such a gating mechanism, blocking mitophagy progression when stabilized on the OMM in a situation of mitochondrial damage. These results provide important mechanistic insight to recent reports that have shown increased levels of Miro1 in cells derived from Parkinson's Disease patients after mitophagy was activated (Hsieh et al., 2019, Hsieh et al., 2016).

Figure 4: H) legend appears to be missing.

We are sorry for the missing legend. We have included this legend in the revised manuscript

Figure 5: Is the upregulation of mitofusins mediated on the transcriptional or only on protein level? Are hyperfused mitochondria a result of enhanced fusion or also downregulation of the fission machinery? Are Opa1 levels affected too?

To address whether mitofusin upregulation occurs at a transcriptional level we have performed quantitative analysis of mitofusin transcript levels from WT and Miro1^{CKO} hippocampus by qPCR. We now show that there is no major change in either mitofusin1 or mitofusin2 transcript levels associated with the increase of their protein levels. We show these data in Supplementary Fig 8A and B. In addition to this we have also performed a ubiquitination assay of mitofusin 2 in Miro1^{KO} neurons and show that there is reduced ubiquitination of mitofusin2 pointing to a reduced degradation of mitofusin by the ubiquitin/proteasome system as the cause for the elevated levels of the mitofusins (Suppl Fig 8C and D). In addition, to address a potential role for downregulation of the fission machinery, we have also performed an analysis on the protein levels of Drp1 and show that they are not altered by the deletion of Miro1 in brains. With this we conclude that the hyperfused mitochondrial phenotype observed in Miro1^{CKO} neurons is due to a dysregulation of mitochondrial dynamics resulting from increased levels of mitofusins.

Is there some compensation between both Miro proteins when one is knocked down/out? It would be important to distinguish between compensatory effects, enhanced mitophagy activation, or reduced turnover.

From our rescue experiments with both Miro1 and Miro2 constructs we conclude that Miro1 is the main regulator of mitophagy initiation and progression. In our experiments Miro2 re-expression in Miro^{DKO} cells did not change the amount of Parkin translocation under mitochondrial damage. In addition, since we do not observe a dramatic remodeling of the mitochondrial network in Miro2^{KO} brains we think that compensation by Miro2 is unlikely to be driving altered mitochondrial homeostasis.

Figure S1: It is unclear what has been measured here. Parkin is (auto)ubiquitinated and moves into higher molecular weight fractions at some point not detectable. Is the same seen in Miro double KO cells?

The graph in Figure S1 (Now Fig 2I) represents the quantification of the non-ubiquitinated ^{Flag}Parkin band shown in the revised Figure 2F. We now clarify that this band is the non-ubiquitinated form of Parkin and that the stabilization of Parkin at three hours supports that the expression of the Miro1^{allR} construct interferes with the normal progression of mitophagy.

References

- Berenguer-Escuder C, Grossmann D, Antony P, Arena G, Wasner K, Massart F, Jarazo J, Walter J, Schwamborn JC, Grunewald A, Kruger R (2020) Impaired mitochondrial-endoplasmic reticulum interaction and mitophagy in Miro1-mutant neurons in Parkinson's disease. *Hum Mol Genet* 29: 1353-1364
- Covill-Cooke C, Toncheva VS, Drew J, Birsa N, Lopez-Domenech G, Kittler JT (2020) Peroxisomal fission is modulated by the mitochondrial Rho-GTPases, Miro1 and Miro2. *EMBO reports* 21: e49865
- Grossmann D, Berenguer-Escuder C, Bellet ME, Scheibner D, Bohler J, Massart F, Rapaport D, Skupin A, Fouquier d'Herouel A, Sharma M, Ghelfi J, Rakovic A, Lichtner P, Antony P, Glaab E, May P, Dimmer KS, Fitzgerald JC, Grunewald A, Kruger R (2019) Mutations in RHOT1 Disrupt Endoplasmic Reticulum-Mitochondria Contact Sites Interfering with Calcium Homeostasis and Mitochondrial Dynamics in Parkinson's Disease. *Antioxid Redox Signal* 31: 1213-1234
- Hsieh CH, Li L, Vanhauwaert R, Nguyen KT, Davis MD, Bu G, Wszolek ZK, Wang X (2019) Miro1 Marks Parkinson's Disease Subset and Miro1 Reducer Rescues Neuron Loss in Parkinson's Models. *Cell Metab* 30: 1131-1140 e7
- Hsieh CH, Shaltouki A, Gonzalez AE, Bettencourt da Cruz A, Burbulla LF, St Lawrence E, Schule B, Krainc D, Palmer TD, Wang X (2016) Functional Impairment in Miro Degradation and Mitophagy Is a Shared Feature in Familial and Sporadic Parkinson's Disease. *Cell Stem Cell* 19: 709-724
- Lopez-Domenech G, Covill-Cooke C, Ivankovic D, Halff EF, Sheehan DF, Norkett R, Birsa N, Kittler JT (2018) Miro proteins coordinate microtubule- and actin-dependent mitochondrial transport and distribution. *The EMBO journal* 37: 321-336
- Modi S, Lopez-Domenech G, Halff EF, Covill-Cooke C, Ivankovic D, Melandri D, Arancibia-Carcamo IL, Burden JJ, Lowe AR, Kittler JT (2019) Miro clusters regulate ER-mitochondria contact sites and link cristae organization to the mitochondrial transport machinery. *Nature communications* 10: 4399
- Puri R, Cheng XT, Lin MY, Huang N, Sheng ZH (2019) Mul1 restrains Parkin-mediated mitophagy in mature neurons by maintaining ER-mitochondrial contacts. *Nature communications* 10: 3645
- Safiulina D, Kuum M, Choubey V, Gogichaishvili N, Liiv J, Hickey MA, Cagalinec M, Mandel M, Zeb A, Liiv M, Kaasik A (2019) Miro proteins prime mitochondria for Parkin translocation and mitophagy. *The EMBO journal* 38

Thank you for submitting your revised study. The manuscript has now been sent back to the original referee #2, who also evaluated the conclusiveness of your responses to referee #1's points.

As you will see, referee #2 finds that the criticisms have been sufficiently addressed and recommends the work for publication. However, this reviewer asks you to improve several points in the discussion and to clarify some others in the rest of the main text.

Our editorial assistants are currently looking at your manuscript. As soon as this process is completed, I will send you a separate e-mail with a list of editorial requests concerning the text and the figures, which should be addressed before we can officially accept your manuscript.

Referee #2:

Overall the authors provide a much improved revised manuscript in which they significantly strengthened and expanded the data. They have adequately addressed the majority of the initial points raised by both referees. The study provides independent confirmation of, but also nicely complements previously published findings. The manuscript provides enough novel aspects and I feel now is a substantial contribution to the field worthy of publication in The EMBO Journal.

However, I do have a few additional suggestions as some aspects could be clarified, discussed a bit more carefully or pointed out as a limitation. Throughout the manuscript, it is also important to clearly specify which of the findings were made in MEFs, in neurons in culture or truly in vivo.

One example is the following sentence that should be more carefully phrased as this is not really shown as stated.

"In summary, in vivo disruption of PINK1/Parkin mediated mitophagy by conditional Miro1 deletion in mouse brain leads to an upregulation of the mitophagic machinery in an age-dependent manner." Conditional Miro1 deletion seems to lead to several effects described herein, while the exact relationship between and the order of events is not fully established yet. In the manuscript it often sounds as if one thing leads to the other, but the individual contributions of altered transport, calcium regulation, and mitophagy impairments to the observed increase in Mfn levels, formation of megamitochondria and hyperactivation of the ISR is not defined. This should be reflected in the discussion and the title. Also the "upregulation" is only convincingly shown for Mfn, not the "mitophagy machinery".

The term 'upregulation of Mfn' could be still be misinterpreted as a transcriptional response, although the authors now show a clear protein effect.

Reviewer's Fig 1. It seems most of the Miro1 mutant constructs show somewhat reduced protein levels, just not (yet) significant due to the error bars. Perhaps this is something that should be discussed as a caveat as protein threshold effect cannot be fully excluded.

With respect to mitophagy measures, I think the authors should be more specific. What they quantify is loss of Tom20 immunoreactivity and not "mitochondrial loss (fraction of cells without any mitochondria after 24 hours of FCCP treatment)" or "complete clearance of mitochondria".

Minor:

Figure 5B: "Flod change" instead of fold change

The following sentence seems odd considering that neither of the constructs is ubiquitinated.
"Although Miro1 K572 ubiquitination by Parkin may be important, and could, for example, impact the rate or kinetics of Miro ubiquitination by Parkin, importantly, we did not find any substantial difference in the ubiquitination kinetics between Miro1R572K (where K572 is the only lysine replaced in the Miro1allR) and Miro1allR."

"Ubiquitinable" might not be the best or most used term.

Point-by-point response to referees' comments:

Referee #2:

Overall the authors provide a much improved revised manuscript in which they significantly strengthened and expanded the data. They have adequately addressed the majority of the initial points raised by both referees. The study provides independent confirmation of, but also nicely complements previously published findings. The manuscript provides enough novel aspects and I feel now is a substantial contribution to the field worthy of publication in The EMBO Journal.

We thank the reviewer for his positive assessment and for supporting publication of our manuscript in The EMBO Journal.

However, I do have a few additional suggestions as some aspects could be clarified, discussed a bit more carefully or pointed out as a limitation. Throughout the manuscript, it is also important to clearly specify which of the findings were made in MEFs, in neurons in culture or truly in vivo.

We have now included comments throughout the manuscript and discussion about where our findings are made. See additions in page 11, 13, 17, and 21 in the revised manuscript.

One example is the following sentence that should be more carefully phrased as this is not really shown as stated. "In summary, in vivo disruption of PINK1/Parkin mediated mitophagy by conditional Miro1 deletion in mouse brain leads to an upregulation of the mitophagic machinery in an age-dependent manner." Conditional Miro1 deletion seems to lead to several effects described herein, while the exact relationship between and the order of events is not fully established yet. In the manuscript it often sounds as if one thing leads to the other, but the individual contributions of altered transport, calcium regulation, and mitophagy impairments to the observed increase in Mfn levels, formation of megamitochondria and hyperactivation of the ISR is not defined. This should be reflected in the discussion and the title. Also, the "upregulation" is only convincingly shown for Mfn, not the "mitophagy machinery".

We have modified the above sentence highlighting the long term consequences of Miro1 deletion in brain, in vivo.

The term 'upregulation of Mfn' could be still be misinterpreted as a transcriptional response, although the authors now show a clear protein effect.

We agree with the referee and have changed "upregulation" to terms that are less associated with a transcriptional activation in abstract, introduction and some places in the discussion.

Reviewer's Fig 1. It seems most of the Miro1 mutant constructs show somewhat reduced protein levels, just not (yet) significant due to the error bars. Perhaps this is something that should be discussed as a caveat as protein threshold effect cannot be fully excluded

In our opinion, the fact that re-expression of both ^{myc}Miro1^{5R} and ^{myc}Miro1^{allR} delay mitochondrial clearance more than re-expression of ^{myc}Miro1^{WT} demonstrates that the expression levels are not an issue. To address this referee's point we have now included the quantification of all the Miro1 ubiquitination mutants shown in Reviewer's Fig 1 in Supplementary information (Now Appendix Figure S2A).

With respect to mitophagy measures, I think the authors should be more specific. What they quantify is loss of Tom20 immunoreactivity and not "mitochondrial loss (fraction of cells without any mitochondria after 24 hours of FCCP treatment)" or "complete clearance of mitochondria".

We have now specified that we quantify Tom20 signal in our experiments of mitochondrial clearance and provided references (results section, page 11; Figure 4 and Figure 4 legend). We have also avoided the use of the word "complete" as we agree that the loss of Tom20 signal, although a good indicator of mitochondrial clearance, may not inform about the "complete absence of mitochondria" (Figure 4 legend).

Minor:

Figure 5B: "Flod change" instead of fold change

We have corrected this mistake

The following sentence seems odd considering that neither of the constructs is ubiquitinated. "Although Miro1 K572 ubiquitination by Parkin may be important, and could, for example, impact the rate or kinetics of Miro ubiquitination by Parkin, importantly, we did not find any substantial difference in the ubiquitination kinetics between Miro1R572K (where K572 is the only lysine replaced in the Miro1allR) and Miro1allR."

We have modified this sentence as follows: "we did not find any substantial ubiquitination of Miro1^{R572K} (where K572 is the only lysine replaced in the Miro1^{allR}) when compared to Miro1^{allR} which cannot be ubiquitinated"

"Ubiquitable" might not be the best or most used term.

We have changed this to the term "ubiquitinatable" which we have seen used in numerous other published papers including from EMBO press.

I am pleased to inform you that your manuscript has been accepted for publication in The EMBO Journal.

YOU MUST COMPLETE ALL CELLS WITH A PINK BACKGROUND ↓
PLEASE NOTE THAT THIS CHECKLIST WILL BE PUBLISHED ALONGSIDE YOUR PAPER

Corresponding Author Name: Josef T. Kittler

Journal Submitted to: The EMBO Journal

Manuscript Number: EMBOJ-2018-100715R